# Evaluation of the smile effect on the Earth Clouds, Aerosols and Radiation Explorer (EarthCARE) /Multi-Spectral Imager (MSI) cloud product

Minrui Wang[1], Takashi Y. Nakajima[1], Woosub Roh[2,3], Masaki Satoh[3], Kentaroh Suzuki[3], Takuji Kubota[4], Mayumi Yoshida[5]

[1]Research & Information Center, Tokai University, Kanagawa, 2591292, Japan
[2]Tokyo University of Marine Science and Technology, Tokyo, 1358533, Japan
[3]Atmosphere and Ocean Research Institute, The University of Tokyo, Chiba, 2778564, Japan
[4]Earth Observation Research Center, Japan Aerospace Exploration Agency, Ibaraki, 3058505, Japan
[5]Remote Sensing Technology Center of Japan, Ibaraki, 3058505, Japan

*Correspondence to*: Minrui Wang (wang.minrui@tokai.ac.jp)

**Abstract.** A cloud identification and profiling algorithm is being developed for the Multi-Spectral Imager (MSI), which is one of the four instruments that the Earth Clouds, Aerosols and Radiation Explorer (EarthCARE) spacecraft will feature. During recent work, we noticed that the MSI response function could shift substantially among some wavelengths (0.67 and 1.65 µm bands) owing to the smile effect, that is an effect in which a shift in the center wavelength appears as a distortion in the spectral image. We evaluated how the smile effect affects the cloud retrieval product qualitatively and quantitatively. We chose four detector pixels from bands 1 and 3 with the nadir pixel as the reference to elucidate how the smile effect error affects the cloud optical thickness ($\tau$) and effective cloud droplet radius ($r_e$) by simulating the MSI forward radiation with Comprehensive Analysis Program for Cloud Optical Measurement (CAPCOM). We also evaluated the error in simulated scenes from a global cloud system resolving model and a satellite simulator to measure the effect on actual observation scenes. For typical shallow warm clouds ($\tau$ = 8, $r_e$ = 8 µm), the smile effect on the cloud retrieval was not significant in most cases (up to 6 % error). For typical deep convective clouds ($\tau$ = 8, $r_e$ = 40 µm), the smile effect on the cloud retrieval was even less significant in most cases (up to 4 % error). Moreover, our results from two oceanic scenes using the synthetic MSI data agreed well with the forward radiation simulation, indicating that the error from the smile effect was generally within 10 %.

## 1 Introduction

Clouds and aerosols are key elements of the Earth's water and energy cycle. Atmospheric radiative forcing is affected by cloud alteration due to indirect aerosol effects. Radiative forcing due to cloud–aerosol interactions still cause the greatest uncertainty in estimating changes in the Earth's energy balance (Solomon et al., 2007). Earth Clouds, Aerosols and Radiation Explorer (EarthCARE) is a joint earth observation satellite project between European Space Agency (ESA) and Japan Aerospace Exploration Agency (JAXA) for observing cloud–aerosol interactions (Illingworth et al. 2015). EarthCARE is equipped with

four sensors, Cloud Profiling Radar (CPR), Atmospheric Lidar (ATLID), Multi-Spectral Imager (MSI), and Broadband Radiometer (BBR). Data products related to clouds, aerosols, and radiation flux are created from single and combined observations from these sensors (Illingworth et al. 2015, Kikuchi et al. 2019).

The MSI (Pérez Albiñana et al. 2010) has been developed by the ESA and measures emitted infrared and reflected solar radiances. The MSI has spectral curvature nonlinearity disturbance, which is known as the smile or frown effect and is a centre wavelength shift that appears as distortions of spectrum images due to misalignment (Yokota et al. 2010; Dadon et al. 2010). The MSI smile effect was reported from ESA in 2017 (Koopman, ed., 2017). Besides, the smile effect has been observed in several previous spaceborne imaging sensors, such as Hyperion Imaging Spectrometer (NASA) (Dadon et al., 2010, Green et al., 2003), Medium Resolution Imaging Spectrometer (MERIS; European Space Agency [ESA]) (ESA, 2008), and Hyper-spectral Imager SUIte (HISUI; Japanese Ministry of Economy, Trade, and Industry) (Japan Space Systems, 2012). The smile effect degrade the spectrum information and reduce classification accuracy, which could cause errors in the cloud retrieval product.

A qualitative and quantitative validation is necessary to evaluate the error caused by the smile effect in the MSI cloud retrievals. First, the effect of the smile effect on the radiation transfer model used in cloud profiling algorithms was evaluated. To evaluate the error in the actual observation scenes, we also used the Joint Simulator for Satellite Sensors (Joint-Simulator) satellite data simulator, which was developed in the JAXA EarthCARE project (Hashino et al., 2013; Satoh et al. 2016; Roh et al., 2020). The satellite data simulator applies satellite orbit calculations and radiation transmission calculations to the cloud/precipitation and temperature/humidity fields generated by the cloud resolving models and general circulation models, and it simulates satellite observations, such as radiances and radar reflectivities. Model verification using pseudo-satellite observation data from a satellite data simulator and actual satellite observation data has been proposed (Matsui et al., 2009, 2016; Masunaga et al., 2010; Roh and Satoh 2014; Roh et al. 2017; Roh et al. 2018). In addition, the pre-launch evaluation of the satellite product using a satellite data simulator has also been conducted (Hagihara et al. 2021; Matsui et al., 2013). The advantage of evaluating algorithms using a satellite data simulator is that satellite data and cloud parameters that are completely time-space matched at all pixels can be obtained.

This paper describes the evaluation of errors caused by the smile effect in the cloud product for EarthCARE MSI observations, especially the microphysical property retrieval of shallow warm cloud and deep convective cloud. The errors are evaluated using algorithms which calculate the MSI standard product in the JAXA (JAXA, 2021). Furthermore, the MSI cloud algorithm to obtain the cloud microphysical property retrieval data is applied to synthetic MSI data, and the retrieval data are compared with and without the smile effect to determine the error. Note that MSI can be used in not only cloud, but also aerosol retrievals, therefore, the smile effect could also affect aerosol products, which is beyond the scope of this paper.

Section 2 describes the cloud product algorithm used in this study and the synthetic MSI data, as well as the methods with which we evaluate the smile effect. Section 3 presents the results and discussion, and in Sect. 4 we give the conclusions of this study.

## 2 Sensors, Data, and Methods

EarthCARE/MSI has the seven bands for cloud remote sensing shown in Table 1 (Pérez Albiñana et al. 2010): two bands in the visible and near infrared region (0.67 and 0.865 µm) for estimating cloud optical thickness (COT, $\tau$); two bands in the short-wave ultra-infrared region (1.65 and 2.21 µm) for estimating cloud particle effective radius (CDR, $r_e$); and three bands in the infrared region (8.8, 10.8, and 12.0 µm) for estimating cloud top temperature and identifying cloud phases. The spatial resolution of each band is 500 m and the swath width is only 150 km. In addition, the swath contains 384 pixels (including 24 dummy pixels on both sides), which is asymmetrical to avoid the sun glint area of the ocean during the local afternoon. When EarthCARE/MSI is in its descending mode (moving from north to south), the nadir pixel is basically located around the 102$^{nd}$ pixel counted from the west. However, in actual observation, the location of nadir will fluctuate slightly according to the location of the satellite, and it is a better way to assign the nadir location from viewing angle than from pixel number. In this study, we defined the location of nadir as the 102$^{nd}$ pixel counted from the west, as a constant value. Based on our definition, the pixel distribution of the MSI swath used in this study is shown in Fig. 1. The smile effect is largest in bands 1 and 3 (Fig. 2), which means that the smile effect could cause errors in both the COT and CDR estimations.

**Table 1: General characteristics of EarthCARE/MSI.**

| Characteristic | Description |
|---|---|
| Instrument | Nadir viewing push-broom imager |
| Spatial resolution | 500 m x 500 m at nadir |
| Swath | 150 km, but -35 km to +115 km<br>(Titled away from sun to minimize sun-glint) |
| Calibration | Sun, on-board warm blackbody, cold space |
| Band | 0.67, 0.865, 1.65, 2.21, 8.8, 10.8, 12.0 µm (band 1 - 7) |

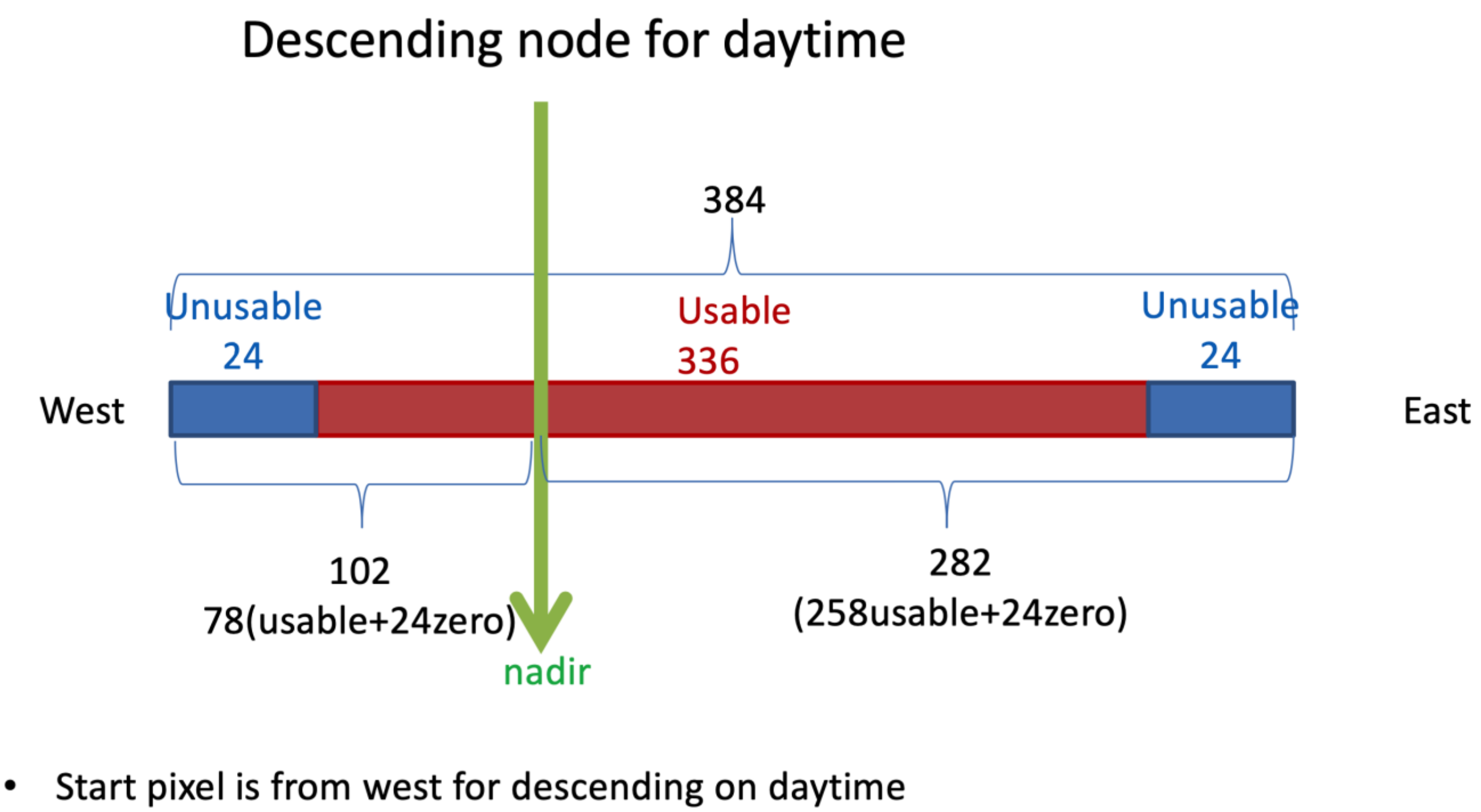

85 **Figure 1: Typical pixel distribution of EarthCARE/MSI swath used in this study. When EarthCARE/MSI is in its descending mode (moving from north to south), the nadir pixel is basically located around the 102nd pixel counted from the west. However, in actual observation, the location of nadir will fluctuate slightly according to the location of the satellite.**

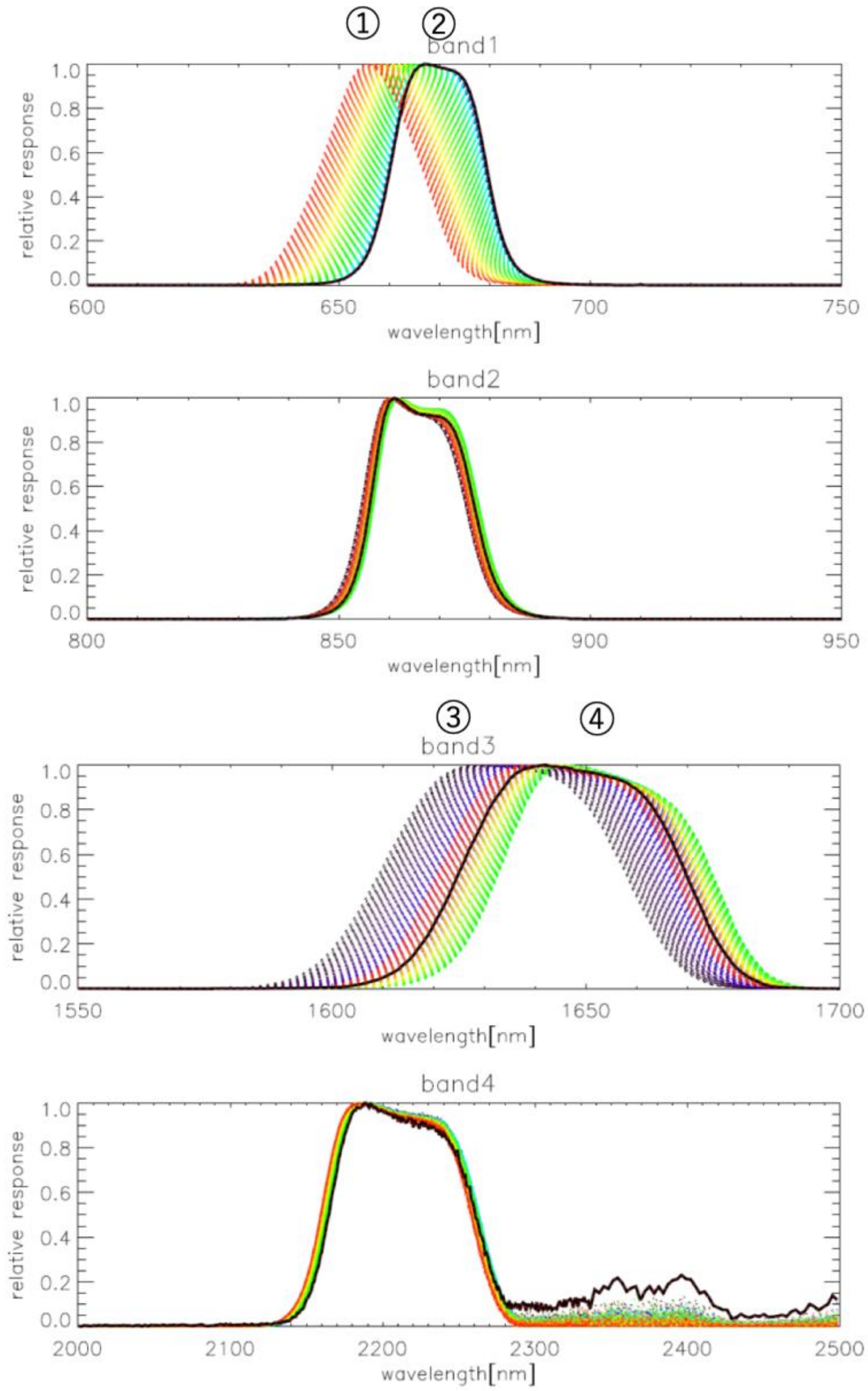

**Figure 2: Wavelength distribution of relative response function on MSI bands 1 to 4 (Value from flight model). The bold line shows the nadir value. Points 1 to 4 show the positions of Pix_BND1_min, Pix_BND1_max, Pix_BND3_min, and Pix_BND3_max, respectively.**

## 2.1 MSI cloud product algorithm

The variables provided by the MSI cloud product contain the cloud flag, cloud phase, COT, CDR, and cloud top temperature. All resolutions are 500 m. The algorithms calculate the MSI standard product in the JAXA consists of cloud flag/cloud phase

algorithm (CLAUDIA) and the Cloud profiling algorithm (CAPCOM). The CLAUDIA is described in Section 2.1.1 and the CAPCOM is described in Section 2.1.2.

### 2.1.1 Cloud flag/cloud phase algorithm (CLAUDIA)

The cloud flag indicates the presence or absence of clouds. The MSI cloud flag algorithm is an optimization of Cloud and Aerosol Unbiased Decision Intellectual Algorithm (CLAUDIA) reported by Ishida and Nakajima (2009) for the MSI observation bands (JAXA, 2021). CLAUDIA expresses the presence or absence of clouds as a real number from 0 (completely cloudy) to 1 (completely sunny), which is called the clear confidence level. During our NICAM/Joint-Simulator data evaluation, we defined two types of cloud: shallow warm cloud (cloud top temperature > 270 K) and deep convective cloud (cloud top temperature < 250 K). We selected two typical scenes for each type of cloud in Sect. 2.2.

### 2.1.2 Cloud profiling algorithm (CAPCOM)

The cloud profiling algorithm, which can also be called the cloud microphysical property retrieval algorithm, is an optimization of Comprehensive Analysis Program for Cloud Optical Measurements (CAPCOM) by Nakajima and Nakajima (1995) and Kawamoto et al. (2001) for the observation bands of MSI. CAPCOM-MSI measures COT, CDR, and cloud top temperature from the observed brightness of the visible (0.67 µm), short wavelength infrared (1.65 or 2.21 µm), and thermal infrared (10.8 µm) bands. In CAPCOM, the observed radiance of the 0.67 µm channel contains information about COT, whereas the observed radiance of the 1.65 and 2.21 µm channels contains information about CDR. At 0.67 µm, the imaginary part of the complex refractive index of water is tiny and hardly influenced by absorption of cloud particles. Therefore, the thicker the cloud optically, the more scattered light travels in the direction of the satellite, and the radiance measured by the satellite increases. In contrast, at wavelengths of 1.65 and 2.21 µm, the imaginary part of the complex refractive index of water is large, so the larger the cloud particle radius, the greater the absorption; thus, the radiance measured by the satellite decreases as the particle size increases. The cloud top altitude and cloud top pressure are estimated from the cloud top temperature using the temperature-altitude or temperature-pressure profile of the objective analysis data. The MSI cloud product provides CDR estimated for bands 3 and 4 (1.65 and 2.21 µm), respectively.

The effective radius of cloud particles, $r_e$, is defined as

$$r_e = \frac{\int_0^\infty r^3 n(r) dr}{\int_0^\infty r^2 n(r) dr}, \tag{1}$$

where $r$ is the particle size and $n(r)$ is the cloud particle number distribution function.

CAPCOM-MSI assumes a lognormal distribution function of the following equation for cloud particle size distribution:

$$n(r) = \frac{c}{r} exp\left[-\frac{(\ln r - \ln r_0)^2}{2\sigma^2}\right]. \tag{2}$$

Here, $c$ is a constant, $r_0$ is the mode radius, and $\sigma$ is the standard deviation of the lognormal distribution. Therefore, the effective radius of cloud particles can be expressed as

$$r_e = r_0 e^{2.5\sigma^2} \,. \tag{3}$$

CAPCOM-MSI assumes $\sigma = 0.35$.

The radiation transfer calculation of CAPCOM-MSI is accelerated by using the look-up table created by the one-dimensional radiation transfer code RSTAR (Nakajima and Tanaka, 1986, 1988; Stamnes et al., 1988). Generally, the response function used for radiation transfer calculation in CAPCOM-MSI is based on the measured value at the nadir location, which was provided by ESA. The bold line in Fig. 2 shows the nadir reference function. We also selected the following four pixels: pix_BND1_min, which gives the response function of the leftmost (shortest wave) in band 1; pix_BND1_max, which gives the response function of the rightmost (longest wave) in band 1; pix_BND3_min, which gives the response function of the leftmost (shortest wave) in band 3; and pix_BND3_max, which gives the response function of the rightmost (longest wave) in band 3. We obtained the response functions at these pixels to evaluate the effects of shifts in the response functions of bands 1 and 3 on the retrieval estimates of cloud microphysical properties. For reference, we also used pix_NADIR, which gives the response function at the nadir pixel. The positions of four selected pixels are shown in Fig. 2 as points 1 to 4. The information for all five pixels is shown in Table 2. We set the solar zenith angle ($\theta$) as 20° or 60° when simulating the radiance.

**Table 2: Pixels selected for smile effect evaluation. The pixel number of the nadir is 102, $\theta_1$ is the satellite zenith angle.**

| | A | B | C | D | E |
|---|---|---|---|---|---|
| | pixel No. | pixel No. from NADIR (\|A - 102\|) | distance from NADIR (B × 0.5) km | tan $\theta_1$ (C / 393) | $\theta_1$ (degree) arctan D |
| pix_BND1_min | 360 | 258 | 129 | 0.32824 | 18.2 |
| pix_BND1_max | 69 | 33 | 17 | 0.04326 | 2.5 |
| pix_BND3_min | 25 | 77 | 39 | 0.09924 | 5.7 |
| pix_BND3_max | 224 | 122 | 61 | 0.15522 | 8.8 |
| pix_NADIR | 102 | 0 | 0 | 0 | 0 |

To evaluate the error caused by the smile effect in the radiation transfer, we used Nakajima-King diagrams. Nakajima-King diagrams were developed for estimating COT and CDR using two wavelength observations in the visible light (e.g., band 1 of MSI) and near-infrared light (e.g., band 3 of MSI) regions (Nakajima and King, 1990; Nakajima et al., 1991). These diagrams are used as the basis of remote sensing of cloud characteristics from visible and near-infrared light observations.

In the EarthCARE/MSI project, cloud characteristic products are divided into standard (water cloud) and research (ice cloud) products. This is because the calculations for water clouds are simpler because the particle shape is roughly spherical, and it is classified as a standard product because it has been analyzed in many projects. In contrast, ice clouds usually have a much greater variety of cloud particle shapes and are classified as research products because they include research elements. Previous research on the EarthCARE/MSI analysis proposed the use of electromagnetic wave scattering solutions using Voronoi-shaped particles (Letu et al., 2016; Letu et al., 2019). Voronoi-shaped particles are also used in the EarthCARE/MSI algorithm for the ice cloud product.

In this study, the combination of COT and CDR to plot Nakajima-King diagrams is defined as follows. For shallow warm clouds, COT (τ) was 1, 2, 4, 8, 16, 32, 48, and 64, and CDR was 2, 4, 8, 16, and 32 μm. For deep convective clouds (Voronoi-shaped particles), COT (τ) was 0.1, 0.5, 1, 2, 4, 8, 16, 32, 48, 64, 80, 90, and 100, and CDR was 5, 10, 20, 40, 60, 80, 100, 110, 120, 130, 140, and 150 μm. We selected the following two combinations of COT and CDR as typical shallow warm clouds and deep convective clouds: for shallow warm clouds, COT was 8 or 32 and CDR was 8 μm; for deep convective clouds, COT was 8 or 32 and CDR was 40 μm.

To evaluate the error of COT and CDR quantitatively, we obtained the first derivation of radiance, $\Delta L$, with respect to COT or CDR from the Nakajima-King diagram as

$\frac{dL}{d\tau}$ and $\frac{dL}{dr_e}$,

and then we obtained the reciprocals as

$\frac{dt}{dL}$ and $\frac{dr_e}{dL}$, respectively,

and multiplied them by the radiance deviation $\Delta L$ between the two response functions (at one of four selected pixels, and on the nadir location) to get the COT and CDR estimation errors, $\Delta\tau$ and $\Delta r_e$, respectively. The results for the error distributions are discussed in Sect. 3.2.

The accuracy requirements for COT and CDR are defined in terms of cloud water content. The liquid water path, *LWP*, of cloud is calculated from the retrieved COT ($\tau_c$) and the effective radius of cloud particles as

$$LWP = \frac{2}{3}\rho\tau_c r_e \,, \tag{4}$$

where $\rho$ is the density of liquid water.

### 2.2 Synthetic MSI data

The synthetic MSI L1 data for MSI cloud product algorithm were created by the Joint-Simulator (Hashino et al., 2013; Satoh et al., 2016), and input 3.5-km-mesh global atmospheric simulation data. The data were calculated by a global storm-resolving atmospheric model, Nonhydrostatic Icosahedral Atmospheric Model (NICAM) (Tomita and Satoh, 2004; Satoh et al., 2008, 2014). Clouds and precipitation in NICAM are computed by the cloud microphysics scheme, the NICAM Single Moment Water 6 (NSW6) (Tomita 2008; Satoh et al. 2014). See previous works (Hashino et al. 2013; Yamada et al. 2016; Nasuno et al. 2016) for the details of the NICAM data. The advantage of the current NICAM simulation data is that it has already been

analyzed in several papers (Hashino et al. 2013, 2016; Matsui et al. 2016; Yamada et al. 2016; Nasuno et al. 2016; Roh et al. 2017, Kubota et al. 2020). The NICAM data were used also in the EarthCARE/CPR doppler simulation (Hagihara et al. 2021). This enables total understanding of the simulation data (in-cluding biases) and appropriate interpretations of the results in developing the satellite data algorithms. The simulation started from 00:00Z 15 June 2008, and the synthetic MSI L1 data is calculated by using the NICAM data on 00:00Z 19 June 2008. We selected two oceanic scenes each for typical shallow warm clouds and typical deep convective clouds, each with 384 pixels in the direction of the swath and 896 pixels in the direction of the track. The geographical location of the shallow warm cloud scenes was 177°–178°W, 22°–25°S, and that for the deep convective cloud scenes was 175°–176°W, 13°–16°S.

There are several sensor simulators in the Joint-Simulator. The R System for Transfer of Atmospheric Radiation—7 (RSTAR7) (Nakajima and Tanaka, 1986, 1988; Nakajima et al., 2003) was used to simulate radiances and brightness temperatures of MSI in the Joint-Simulator. The size distributions for cloud ice and cloud water are not defined by the cloud microphysics scheme in NICAM (NSW6). Therefore, effective radii of 40 and 8 µm were used for the cloud ice and cloud water in the Joint-Simulator, respectively. Single scatterings of hydrometeors as spherical shapes were calculated with Mie theory in the Joint-Simulator. We interpolated the response function using 60 bins for each channel. The smile effect was considered using response functions depending on the pixel number from ESA (Fig. 2). MSI data were generated using the fixed response function at the nadir location as the control data. By applying the MSI algorithm to the nadir and smile effect sets of simulated radiance data, we compared the two sets of cloud retrieval product and evaluated the error caused by the smile effect.

### 2.3 Evaluation criteria for the smile effect error

As our evaluation criteria for the smile effect error, we performed the following analytical estimation of shortwave flux ($F_{sw}$) due to $r_e$ change under the assumed constant cloud water content, $W$.

From the formulation of $F_{sw}$,

$$F_{SW} = -\frac{S_0 n e^{-0.2} \Delta\alpha}{4}, \tag{5}$$

we get the relationship between the change of albedo ($\Delta\alpha$) and COT ($\Delta\tau$),

$$\alpha = \frac{(1-g)\tau}{1+(1-g)\tau}, \tag{6}$$

$$\Delta\alpha = \frac{\alpha(1-\alpha)\Delta\tau}{\tau}, \tag{7}$$

$$\frac{\Delta\alpha}{\alpha} = \frac{(1-\alpha)\Delta\tau}{\tau}. \tag{8}$$

Meanwhile,

$$\tau = \frac{kW}{r_e}, \tag{9}$$

$$\Delta\tau = k\left(\frac{\Delta W}{r_e} - \frac{W}{{r_e}^2 \Delta r_e}\right) = \frac{\Delta W}{W\tau} - \frac{\Delta r_e}{r_e \tau}, \tag{10}$$

$$\frac{\Delta\tau}{\tau} = \frac{\Delta W}{W} - \frac{\Delta r_e}{r_e}, \tag{11}$$

assuming that W is constant ($\Delta W = 0$), and

$$\frac{\Delta\alpha}{\alpha} = -\frac{(1-\alpha)\Delta r_e}{r_e}, \quad (12)$$

and according to Eqs. (5), (6), and (12), we assumed that global mean optical thickness $\tau = 7$ and g = 0.85, and thus from Eq. (6) we get $\alpha = 0.51$. Then Eq. (5) becomes

$$F_{SW} = 343 \times 0.6 \times 0.81 \times 0.51 \times (1-0.51) \times \frac{\Delta r_e}{r_e} = 42 \times \frac{\Delta r_e}{r_e}. \quad (13)$$

According to Eq. (13), under the global mean distribution, if CDR decreased by 10 %, then $F_{sw}$ would decrease by about 4.2 $W/m^2$. An error of this size or larger would be non-negligible in the cloud profiling algorithm of EarthCARE/MSI. Therefore, we focused on every $\Delta\tau$ and $\Delta r_e$ result to ensure that in most cases, the error caused by the smile effect did not exceed this value.

## 3 Results and Discussion

### 3.1 Smile effect in radiation transfer simulation (Nakajima-King diagrams)

The Nakajima-King diagrams for shallow warm clouds are shown in Figs. 3 (solar zenith angle $\theta_0 = 60°$) and 4 ($\theta_0 = 20°$). The red lines show the results obtained using the response function with the smile effect in Pix_BND1_min, Pix_BND1_max, Pix_BND3_min, and Pix_BND3_max. The black lines show the results obtained using the response function located at the nadir pixel, which is not affected by the smile effect. The satellite zenith angle for each pixel is shown in Table 2. In pixels in band 1 (panels (a) and (b)), the COT error ($\Delta\tau$) is much larger than the CDR error ($\Delta r_e$), whereas the opposite is observed in pixels in band 3 (panels (c) and (d)). This is because the observed radiance of the 0.67 µm channel mainly contains information about COT, whereas the observed radiance of the 1.65 and 2.21 µm channels contains information about CDR, as we mentioned in session 2.1.2.

225

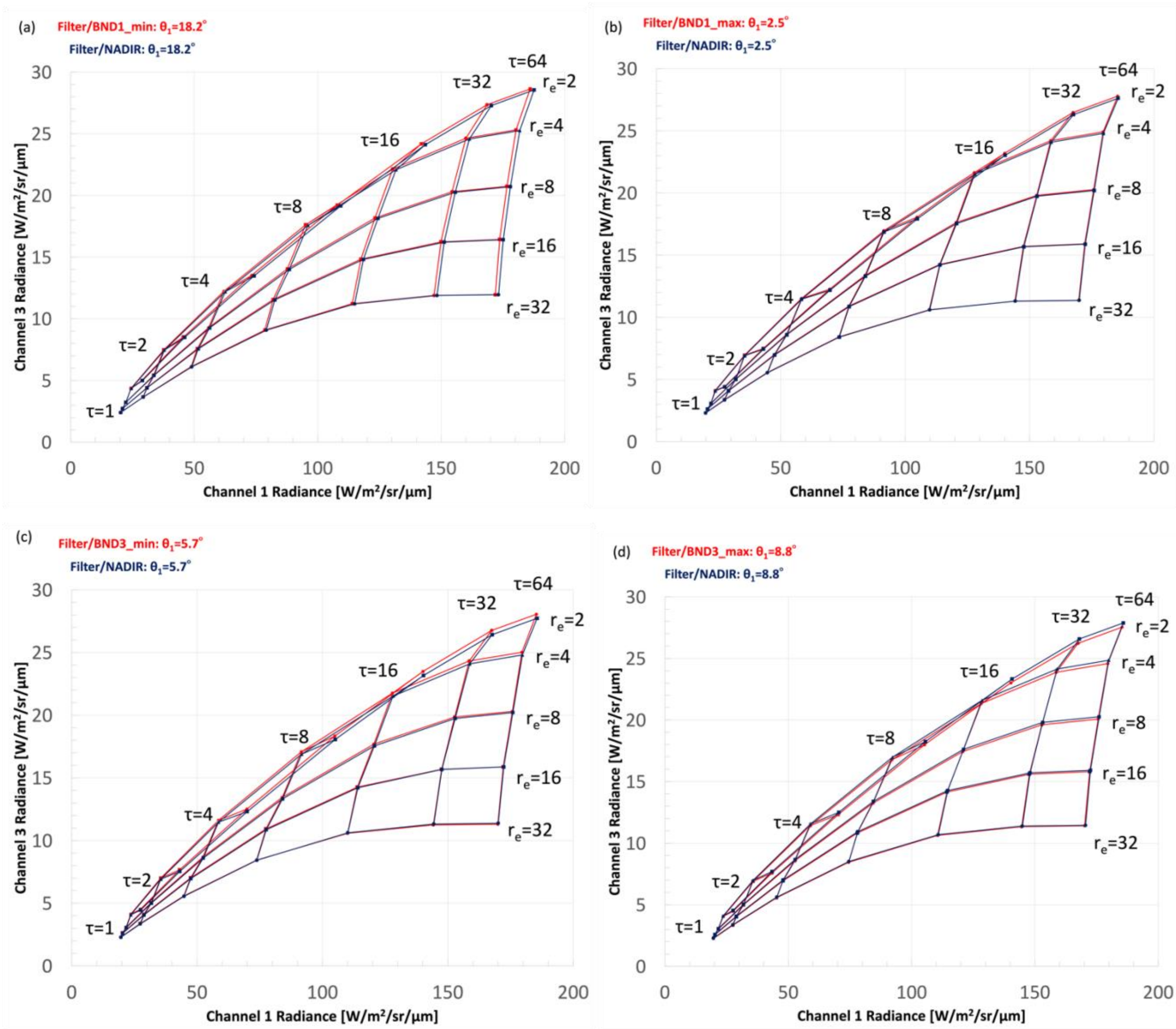

**Figure 3: Nakajima-King diagrams for shallow warm clouds at (a) pix_BND1_min, (b) pix_BND1_max, (c) pix_BND3_min, and (d) pix_BND3_max for a solar zenith angle of 60°. The red lines show the results from the response function with the smile effect. The black lines show the results from the nadir response function.**

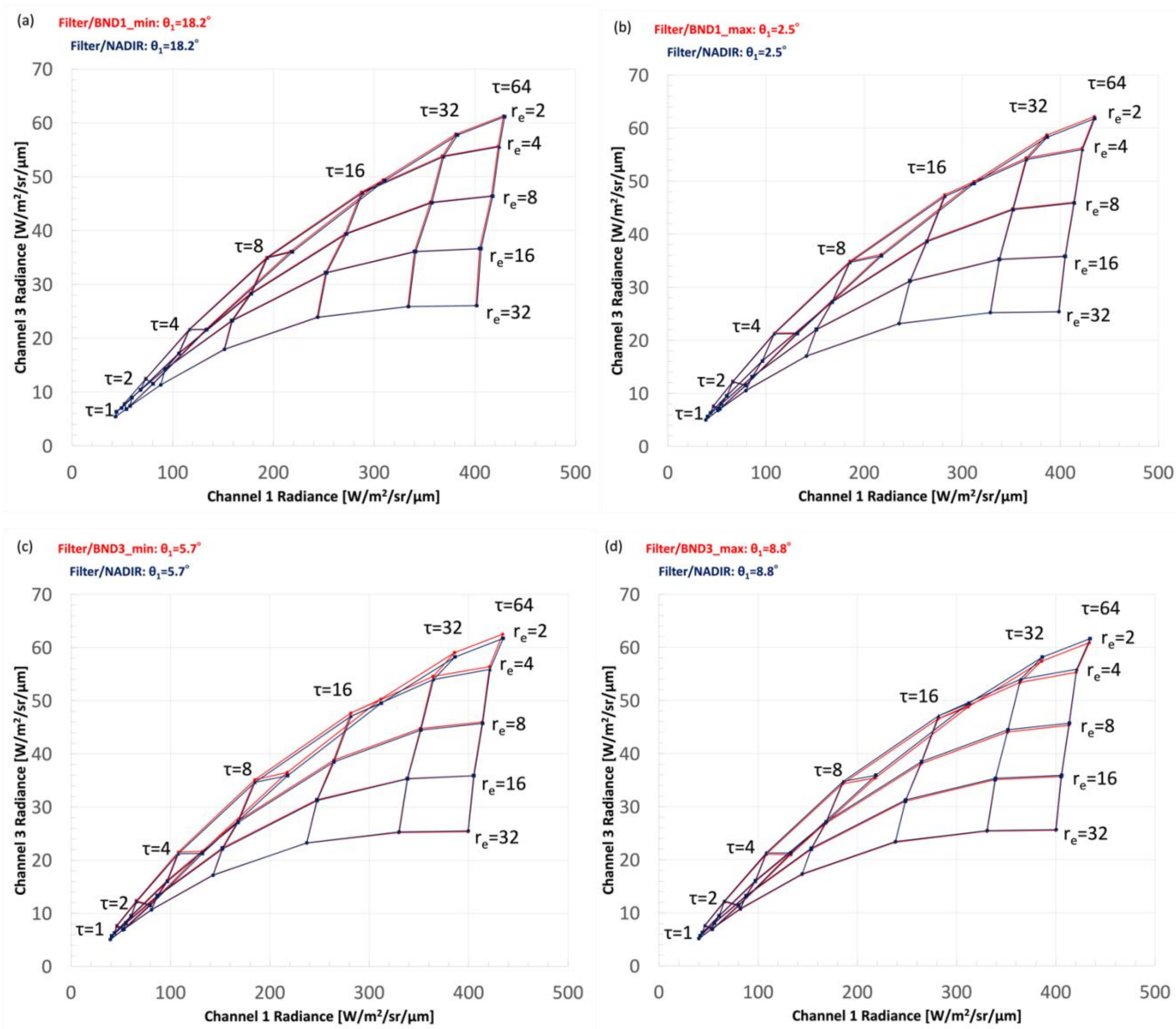

**Figure 4: Nakajima-King diagrams for shallow warm clouds at (a) pix_BND1_min, (b) pix_BND1_max, (c) pix_BND3_min, and (d) pix_BND3_max for a solar zenith angle of 20°. The red lines show the results from the response function with the smile effect. The black lines show the results from the nadir response function.**

The Nakajima-King diagrams for deep convective clouds are shown in Figs. 5 ($\theta$ = 60°) and 6 ($\theta$ = 20°). The same trends for $\Delta\tau$ and $\Delta r_e$ as for shallow warm clouds are seen for deep convective clouds. Although we had wide ranges for COT (up to 100) and CDR (up to 150 μm) in our simulation, these extreme values do not exist in general in ice cloud research products because such large COT and CDR values usually occur for no ice clouds.

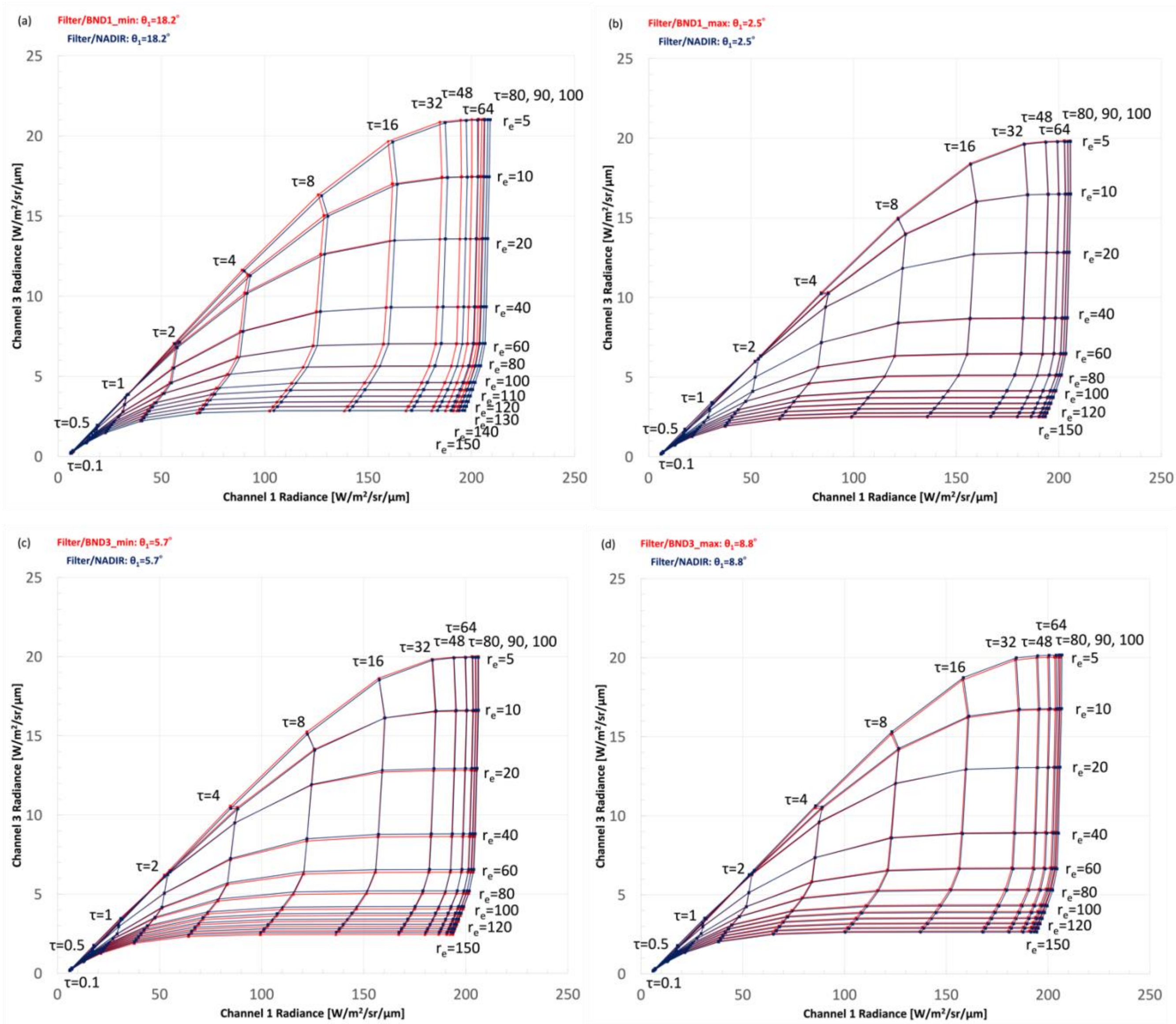

**Figure 5: Nakajima-King diagrams for deep convective clouds at (a) pix_BND1_min, (b) pix_BND1_max, (c) pix_BND3_min, and (d) pix_BND3_max for a solar zenith angle of 60°. The red lines show the results from the response function with the smile effect. The black lines show the results from the nadir response function.**

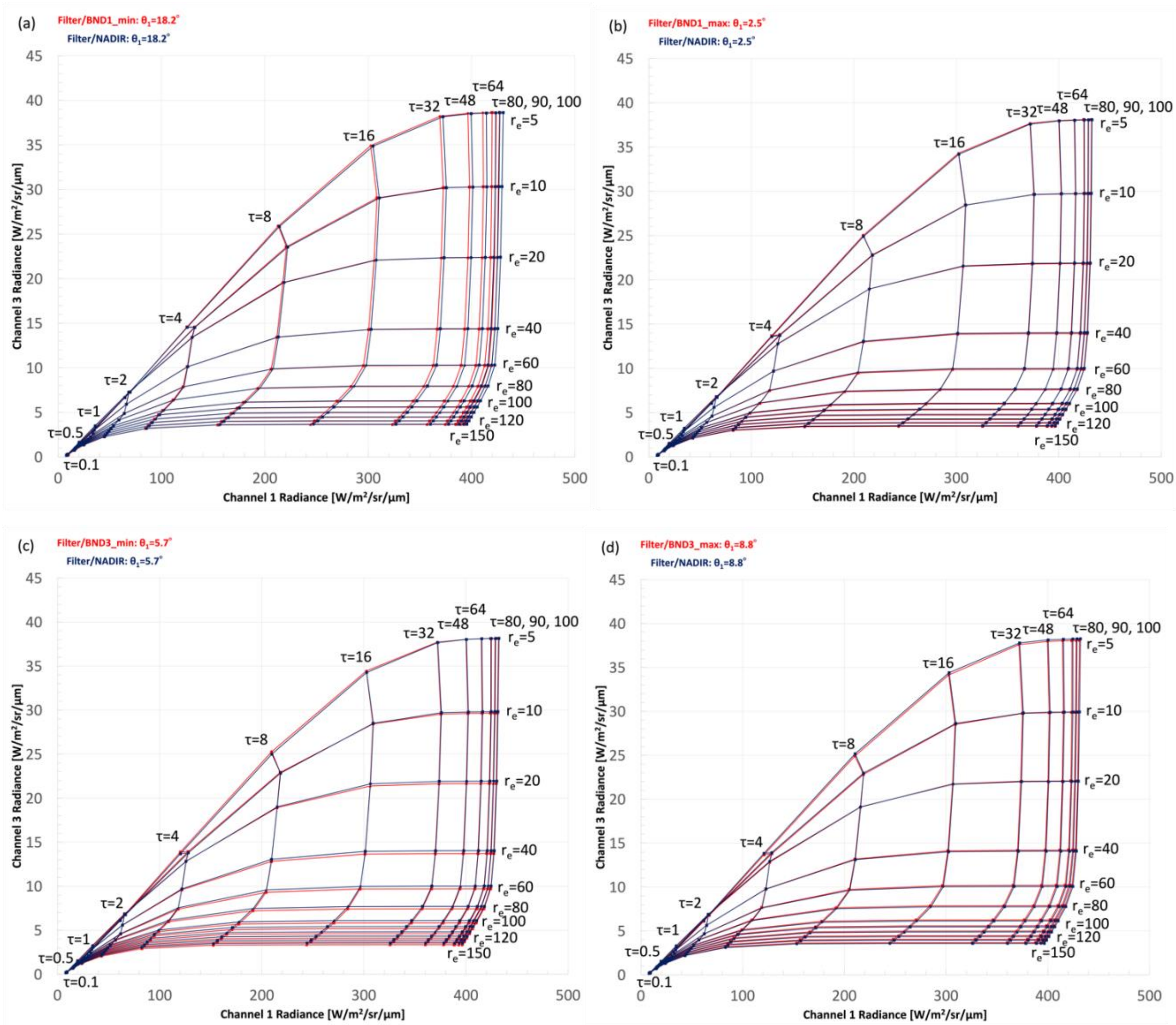

**Figure 6: Nakajima-King diagrams for deep convective clouds at (a) pix_BND1_min, (b) pix_BND1_max, (c) pix_BND3_min, and (d) pix_BND3_max for a solar zenith angle of 20°. The red lines show the results from the response function with the smile effect. The black lines show the results from the nadir response function.**

Based on the Nakajima-King diagrams, we calculated $\Delta\tau$ and $\Delta r_e$ for typical shallow warm cloud and deep convective cloud cases and the results are shown in Tables 3 and 4. The trends in $\Delta\tau$ and $\Delta r_e$ which we mentioned in session 2.1.2 were also observed here, but $\Delta\tau$ and $\Delta r_e$ did not exceed our 10 % evaluation criteria.

**Table 3: Errors of COT and CDR for typical shallow warm cloud ($\tau$ = 8 or 32, $r_e$ = 8 µm).**

| | | $\tau$=8, $r_e$=8 μm | | $\tau$=32, $r_e$=8 μm | |
|---|---|---|---|---|---|
| | | $\Delta\tau$ | $\Delta r_e$(μm) | $\Delta\tau$ | $\Delta r_e$(μm) |
| $\theta_0$=60° SWC | pix_BND1_min | 0.19(2.3%) | 0.13(1.6%) | 0.89(2.7%) | 0.08(1.0%) |
| | pix_BND1_max | 0.02(0.2%) | 0.23(2.8%) | 0.11(0.3%) | 0.14(1.7%) |
| | pix_BND3_min | 0.03(0.3%) | 0.38(4.8%) | 0.13(0.4%) | 0.21(2.6%) |
| | pix_BND3_max | 0.05(0.6%) | 0.50(6.2%) | 0.24(0.7%) | 0.38(4.7%) |
| $\theta_0$=20° SWC | pix_BND1_min | 0.08(1.0%) | 0.11(1.4%) | 0.16(0.5%) | 0.07(0.9%) |
| | pix_BND1_max | 0.008(0.1%) | 0.23(2.9%) | 0.05(0.2%) | 0.14(1.8%) |
| | pix_BND3_min | 0.02(0.2%) | 0.46(5.7%) | 0.05(0.2%) | 0.25(3.1%) |
| | pix_BND3_max | 0.03(0.3%) | 0.51(6.3%) | 0.11(0.3%) | 0.37(4.6%) |

※$\theta_0$ = solar zenith angle, $\tau$ = optical thickness, $r_e$ = effective radius of cloud droplet

**Table 4: Errors of COT and CDR for typical deep convective cloud ($\tau$ = 8 or 32, $r_e$ = 40 μm).**

| | | $\tau$=8, $r_e$=40 μm | | $\tau$=32, $r_e$=40 μm | |
|---|---|---|---|---|---|
| | | $\Delta\tau$ | $\Delta r_e$(μm) | $\Delta\tau$ | $\Delta r_e$(μm) |
| $\theta_0$=60° DCC | pix_BND1_min | 0.176(2.2%) | 0.113(0.3%) | 1.120(3.5%) | 0.148(0.4%) |
| | pix_BND1_max | 0.011(0.1%) | 0.348(0.9%) | 0.062(0.2%) | 0.411(1.0%) |
| | pix_BND3_min | 0.014(0.2%) | 1.310(3.3%) | 0.095(0.3%) | 1.470(3.7%) |
| | pix_BND3_max | 0.035(0.4%) | 0.282(0.7%) | 0.220(0.7%) | 0.394(1.0%) |
| $\theta_0$=20° DCC | pix_BND1_min | 0.020(0.2%) | 0.168(0.4%) | 0.103(0.3%) | 0.196(0.5%) |
| | pix_BND1_max | 0.003(0.04%) | 0.398(1.0%) | 0.023(0.1%) | 0.497(1.2%) |
| | pix_BND3_min | 0.004(0.1%) | 1.370(3.4%) | 0.024(0.1%) | 1.650(4.1%) |
| | pix_BND3_max | 0.004(0.1%) | 0.405(1.0%) | 0.030(0.1%) | 0.550(1.4%) |

※$\theta_0$ = solar zenith angle, $\tau$ = optical thickness, $r_e$ = effective radius of cloud droplet

### 3.2 Error distribution in radiation transfer

The error distributions for every combination of COT and CDR mentioned in Sect. 2.1.2 are shown in Figs. 7–10 for the shallow warm clouds and Figs. 11–14 for the deep convective clouds at Pix_BND1_min, Pix_BND1_max, Pix_BND3_min, and Pix_BND3_max. The top two panels in every figure show the results for a solar zenith angle of 20°, and the bottom two panels show the results for a solar zenith angle of 60°. The left two panels in every figure show the distributions of $\Delta r_e$, and the right two panels show the distributions of $\Delta\tau$. The red frames show the position of a typical shallow warm clouds (COT = 8 or 32, CDR = 8 μm) or deep convective clouds (COT = 8 or 32, CDR = 40 μm) in each panel.

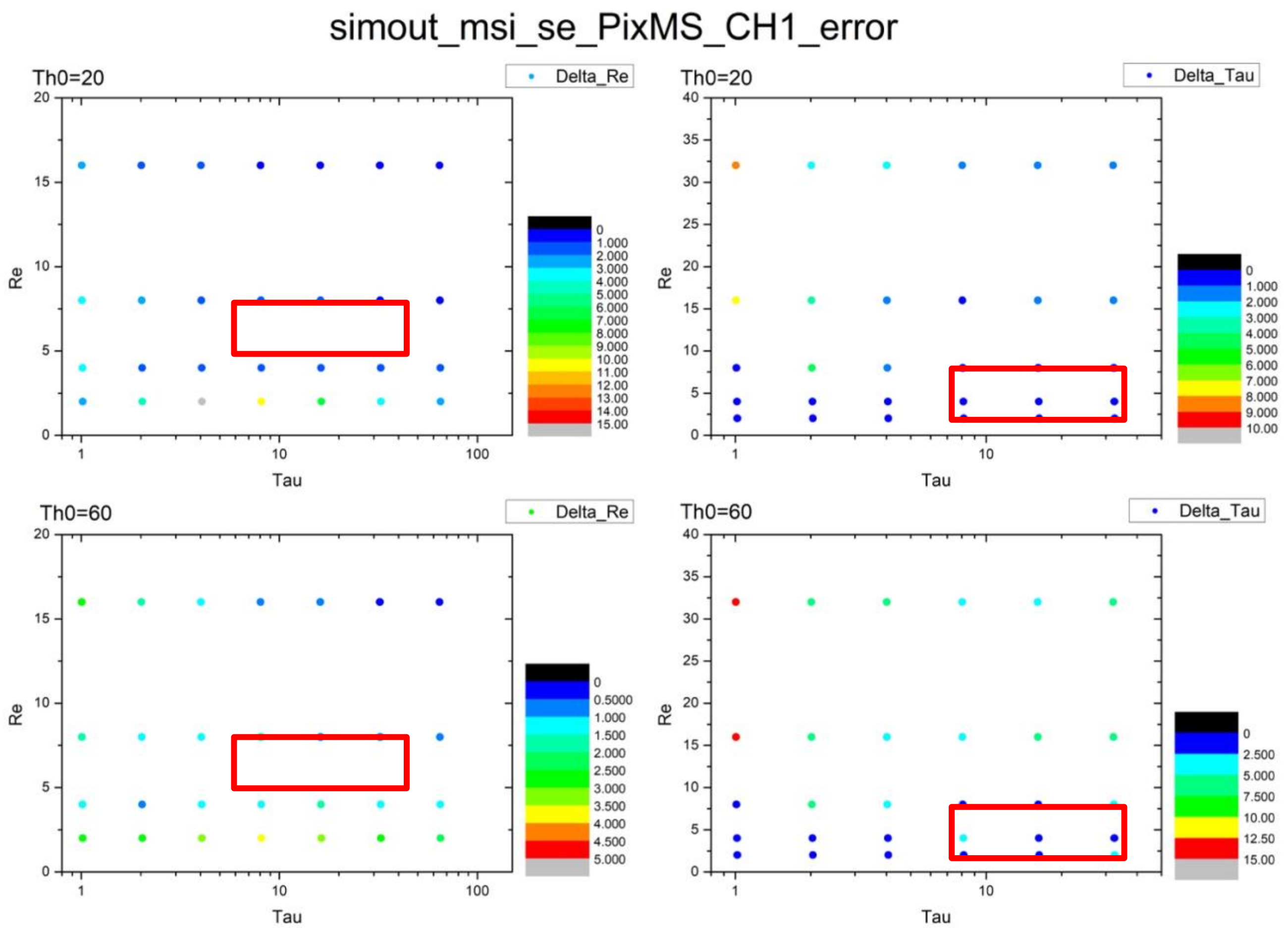

**Figure 7: Error distributions of COT and CDR from the Nakajima-King diagram for shallow warm clouds at pix_BND1_min. Upper left panel: distribution of $\Delta r_e$ for a solar zenith angle of 20°. Upper right panel: distribution of $\Delta\tau$ for a solar zenith angle of 20°. Lower left panel: distribution of $\Delta r_e$ for a solar zenith angle of 60°. Lower right panel: distribution of $\Delta\tau$ for a solar zenith angle of 60°. The red frames indicate the presence of typical shallow warm cloud. COT = 8 or 32, CDR = 8 µm.**

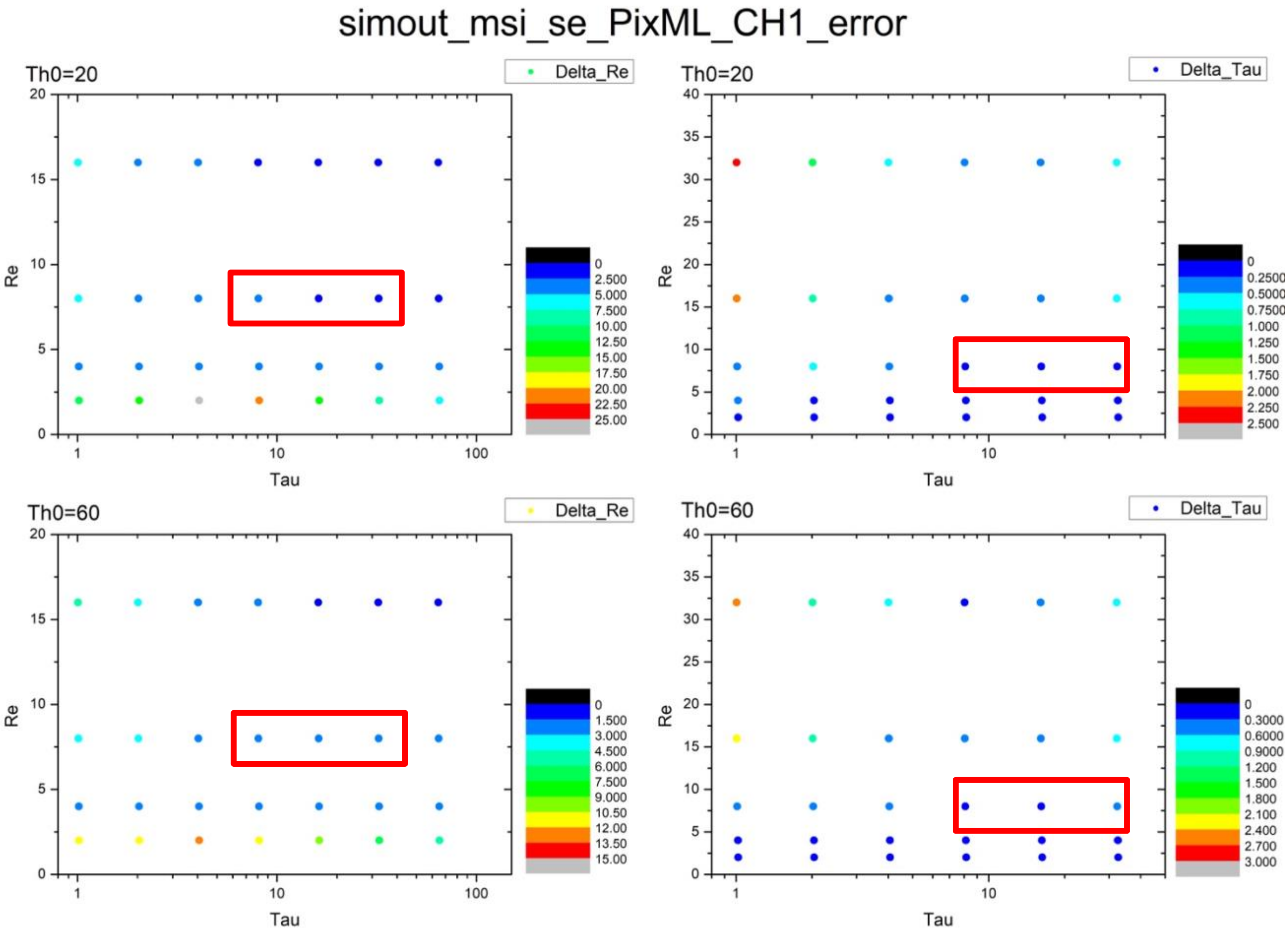

270 **Figure 8: Error distributions of COT and CDR from the Nakajima-King diagram for shallow warm clouds at pix_BND1_max. Upper left panel: distribution of $\Delta r_e$ for a solar zenith angle of 20°. Upper right panel: distribution of $\Delta\tau$ for a solar zenith angle of 20°. Lower left panel: distribution of $\Delta r_e$ for a solar zenith angle of 60°. Lower right panel: distribution of $\Delta\tau$ for a solar zenith angle of 60°. The red frames show the presence of typical shallow warm cloud. COT = 8 or 32, CDR = 8 μm.**

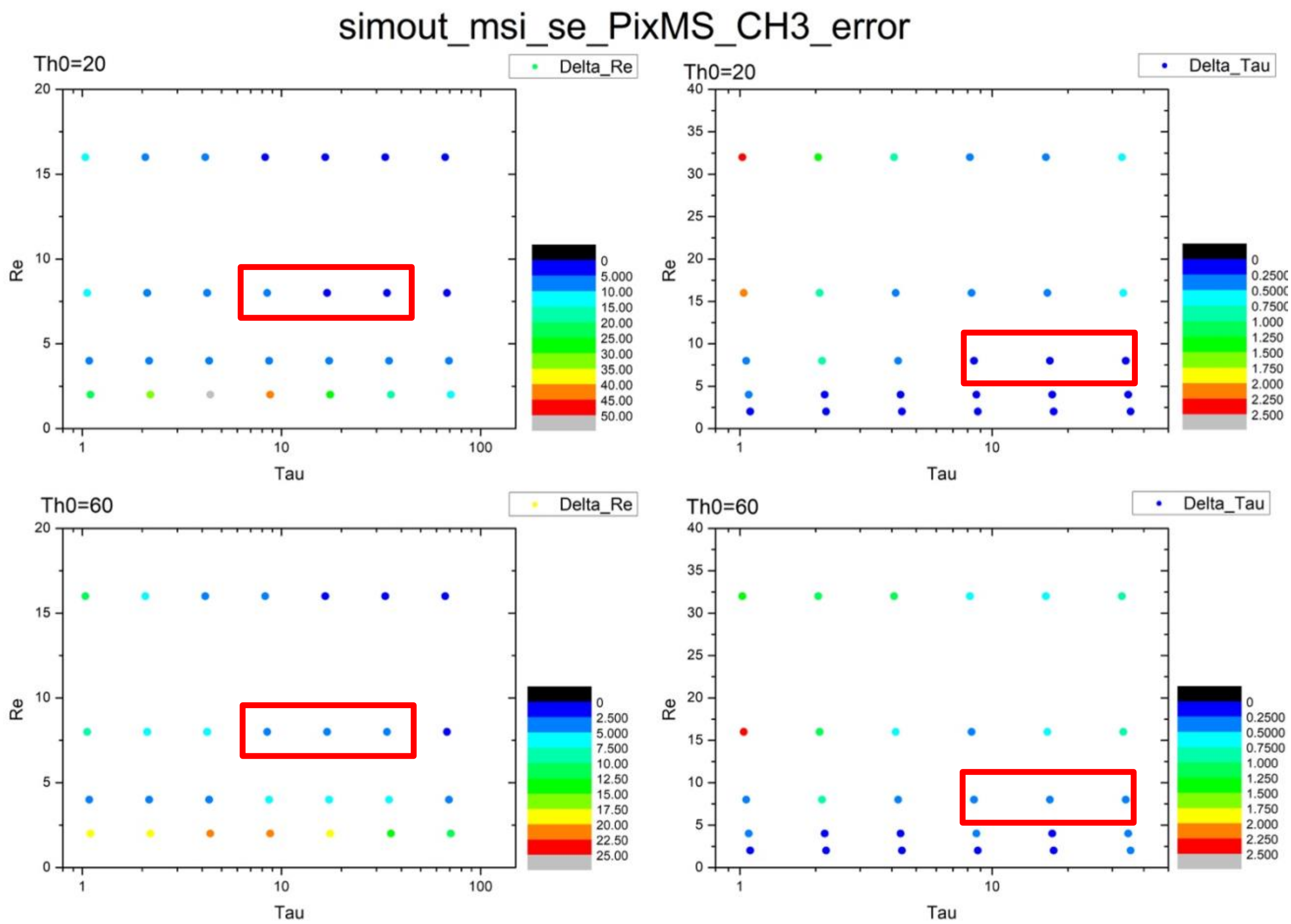

**Figure 9: Error distributions of COT and CDR from the Nakajima-King diagram for shallow warm clouds at pix_BND3_min. Upper left panel: distribution of $\Delta r_e$ for a solar zenith angle of 20°. Upper right panel: distribution of $\Delta\tau$ for a solar zenith angle of 20°. Lower left panel: distribution of $\Delta r_e$ for a solar zenith angle of 60°. Lower right panel: distribution of $\Delta\tau$ for a solar zenith angle of 60°. The red frames show the presence of typical shallow warm cloud. COT = 8 or 32, CDR = 8 µm.**

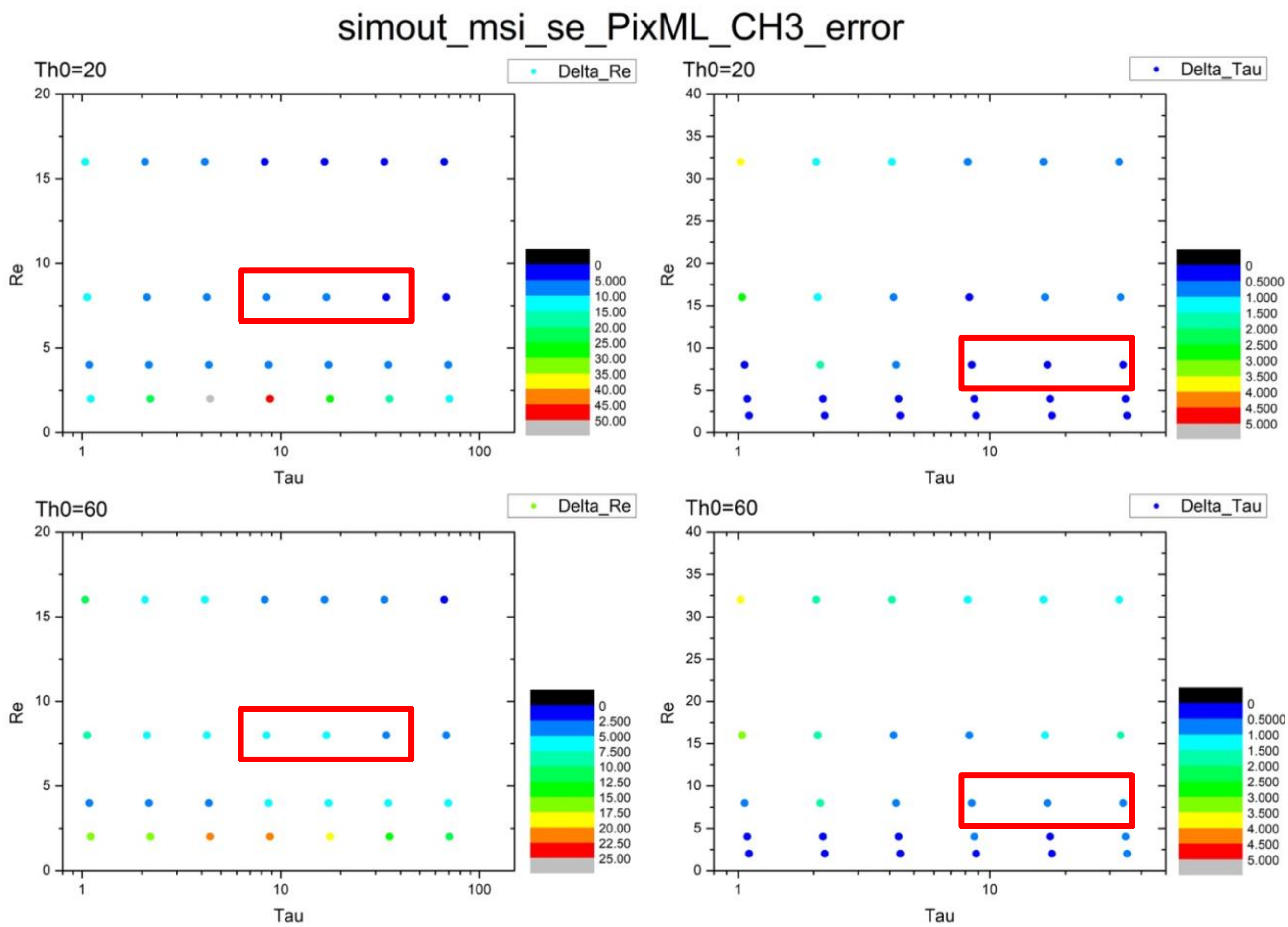

**Figure 10: Error distributions of COT and CDR from the Nakajima-King diagram for shallow warm clouds at pix_BND3_max. Upper left panel: distribution of $\Delta r_e$ for a solar zenith angle of 20°. Upper right panel: distribution of $\Delta\tau$ for a solar zenith angle of 20°. Lower left panel: distribution of $\Delta r_e$ for a solar zenith angle of 60°. Lower right panel: distribution of $\Delta\tau$ for a solar zenith angle of 60°. The red frames show the presence of typical shallow warm clouds. COT = 8 or 32, CDR = 8 µm.**

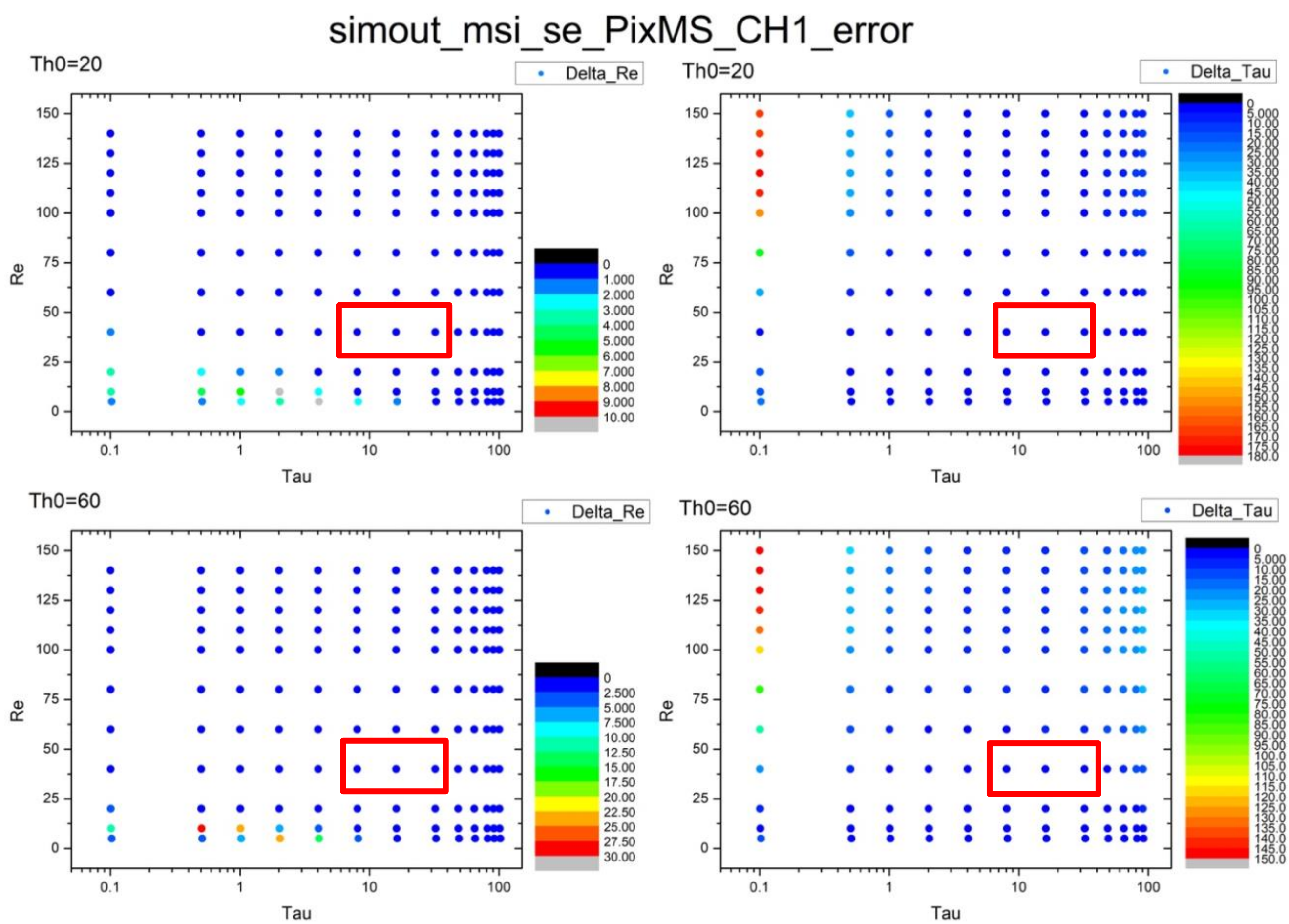

**Figure 11: Error distributions of COT and CDR from the Nakajima-King diagram for deep convective clouds at pix_BND1_min. Upper left panel: distribution of $\Delta r_e$ for a solar zenith angle of 20°. Upper right panel: distribution of $\Delta\tau$ for a solar zenith angle of 20°. Lower left panel: distribution of $\Delta r_e$ for a solar zenith angle of 60°. Lower right panel: distribution of $\Delta\tau$ for a solar zenith angle of 60°. The red frames show the presence of typical deep convective clouds. COT = 8 or 32, CDR = 40 μm.**

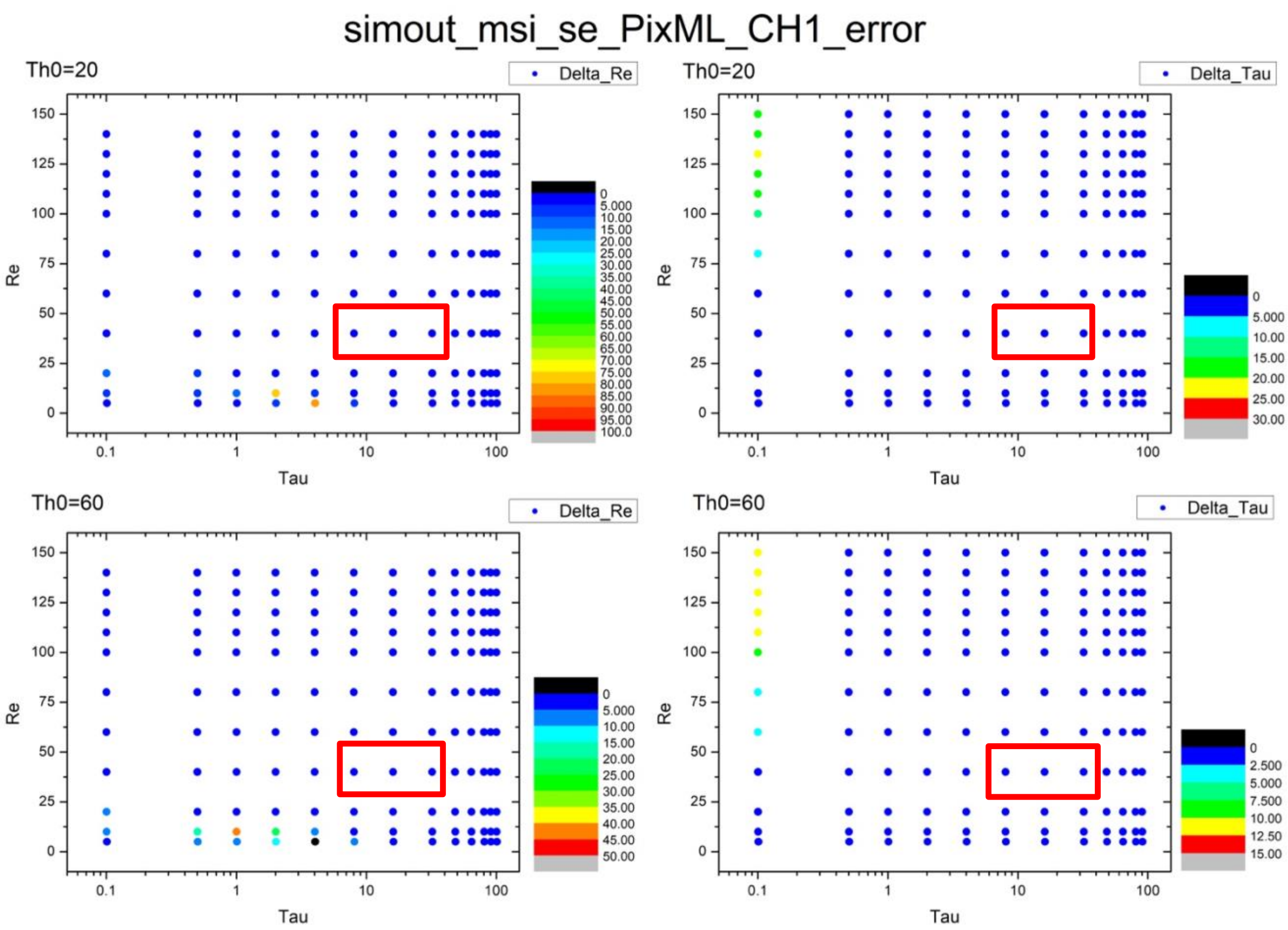

**Figure 12: Error distributions of COT and CDR from the Nakajima-King diagram for deep convective clouds at pix_BND1_max. Upper left panel: distribution of $\Delta r_e$ for a solar zenith angle of 20°. Upper right panel: distribution of $\Delta\tau$ for a solar zenith angle of 20°. Lower left panel: distribution of $\Delta r_e$ for a solar zenith angle of 60°. Lower right panel: distribution of $\Delta\tau$ for a solar zenith angle of 60°. The red frames show the presence of typical deep convective clouds. COT = 8 or 32, CDR = 40 μm.**

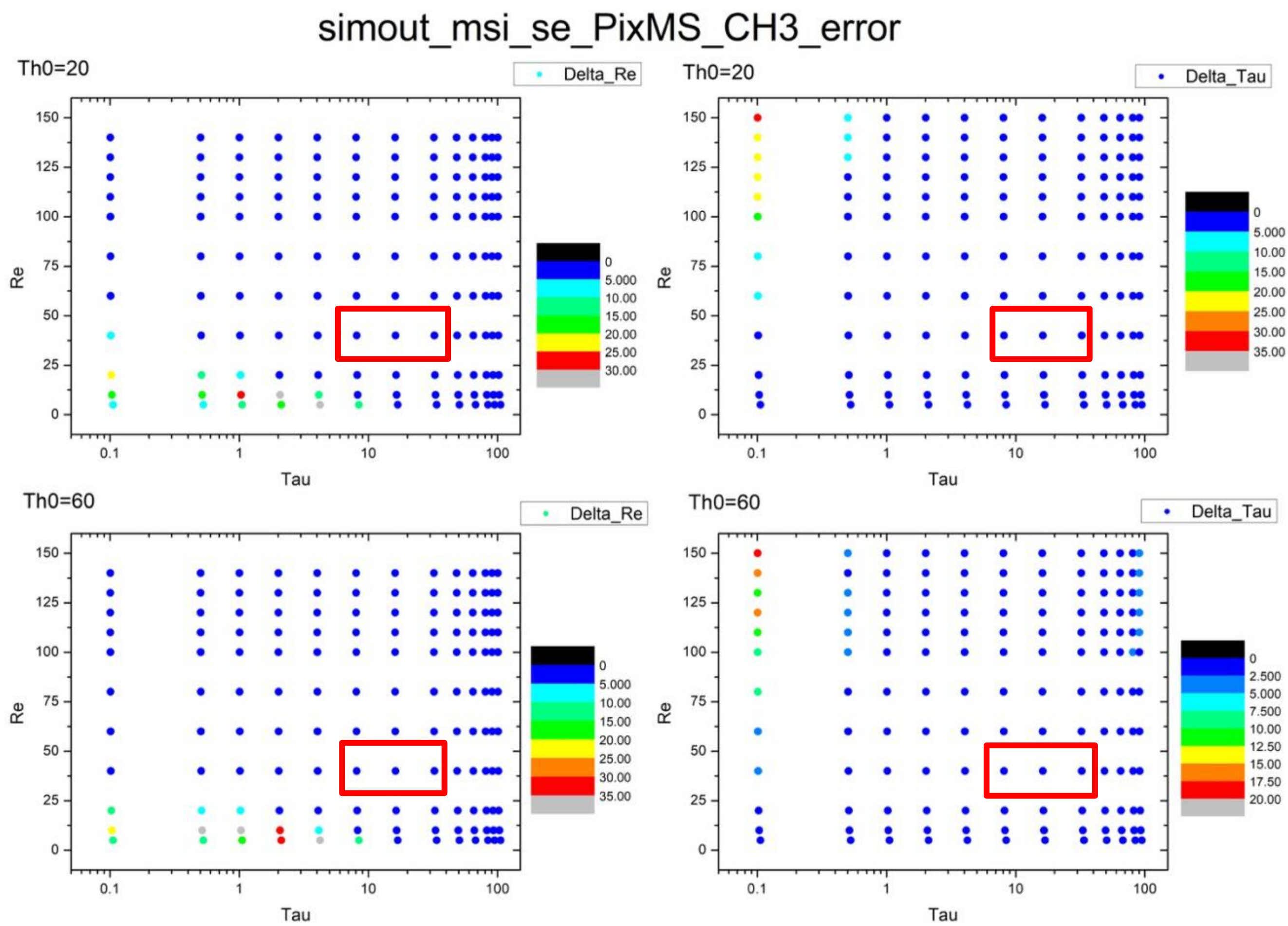

300 **Figure 13: Error distribution of COT and CDR from Nakajima-King diagram for deep convective clouds, at pix_BND3_min. Upper left panel: distribution of $\Delta r_e$ for a solar zenith angle of 20°. Upper right panel: distribution of $\Delta\tau$ for a solar zenith angle of 20°. Lower left panel: distribution of $\Delta r_e$ for a solar zenith angle of 60°. Lower right panel: distribution of $\Delta\tau$ for a solar zenith angle of 60°. The red frames show the presence of typical deep convective clouds. COT = 8 or 32, CDR = 40 μm.**

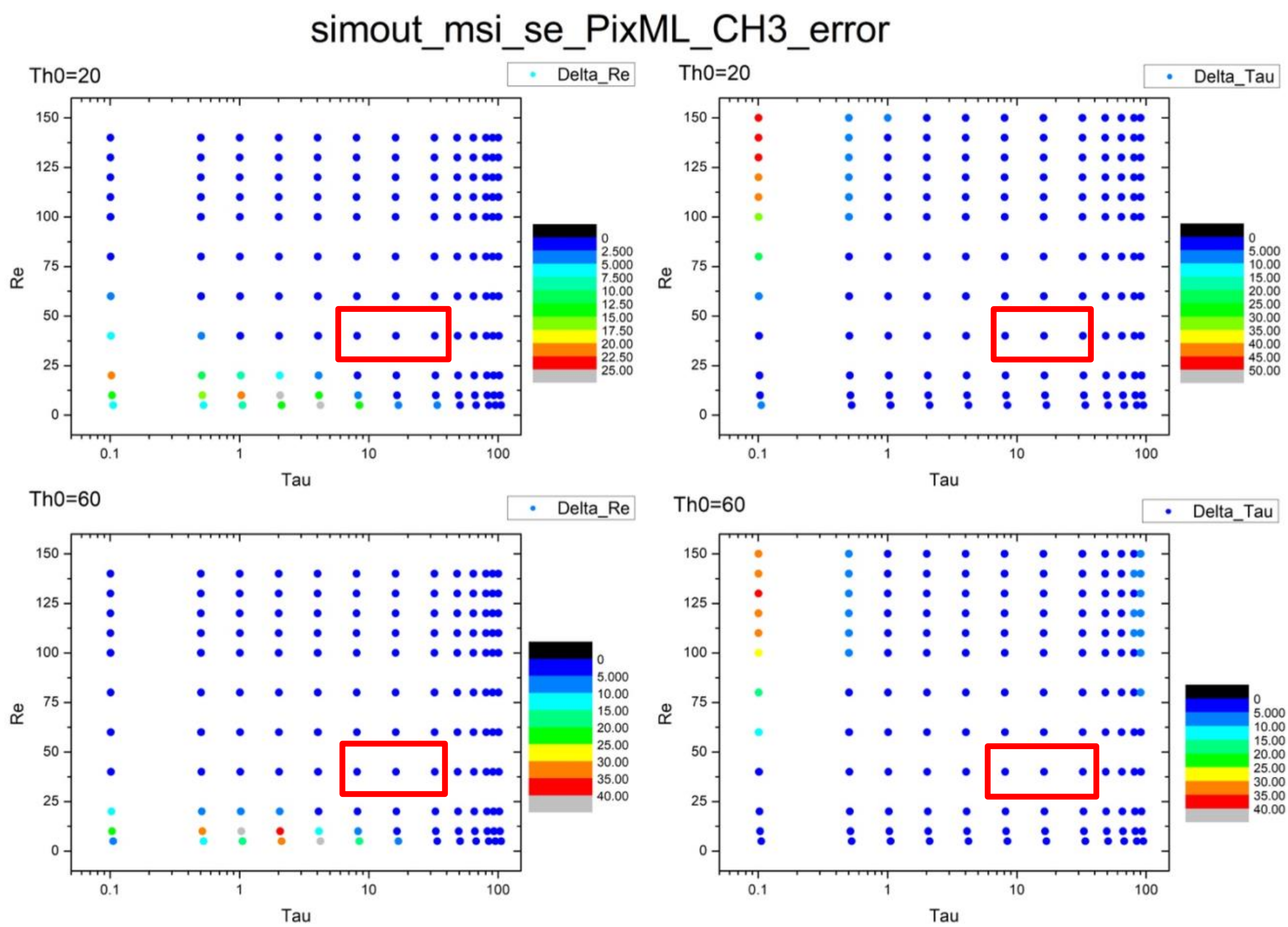

**Figure 14: Error distributions of COT and CDR from the Nakajima-King diagram for deep convective clouds at pix_BND3_max. Upper left panel: distribution of $\Delta r_e$ for a solar zenith angle of 20°. Upper right panel: distribution of $\Delta\tau$ for a solar zenith angle of 20°. Lower left panel: distribution of $\Delta r_e$ for a solar zenith angle of 60°. Lower right panel: distribution of $\Delta\tau$ for a solar zenith angle of 60°. The red frames show the presence of typical deep convective clouds. COT = 8 or 32, CDR = 40 µm.**

According to all 32 panels in Figs. 7–14, none of the $\Delta\tau$ and $\Delta r_e$ values of typical clouds exceed our 10 % evaluation criteria, which are also shown in Tables 3 and 4. $\Delta\tau$ and $\Delta r_e$ are high when low COT ($\tau$ = 1 or 0.1) and CDR ($r_e$ = 2 or 5 µm) values are used in the calculation, as shown by the red-orange dots in the bottom-left part of some $\Delta r_e$ panels and in the top-left part of some $\Delta\tau$ panels. Especially in Fig. 11 (Pix_BND1_min for deep convective cloud case), extreme values of $\Delta\tau$ can even exceed 100 %. This is because the derivative of radiance, $\Delta L$, is much larger during the radiation transfer simulation with COT and CDR values that are too low (COT < 1 for both clouds, CDR < 3 µm for shallow warm cloud, and CDR < 10 µm for deep

convective cloud), and generally very low COT and CDR values are rare for cloud properties. Thus, none of these high error results have a definitive meaning and are neglectable in the smile effect error evaluation. Similarly, points with very high CDR (>100 µm) and very low COT (0.1) in deep convective cloud panels are also unrealistic for cloud properties, which means these cases are also neglectable during our evaluation, regardless of how large the error is.

### 3.3 Error evaluation in NICAM/Joint-Simulator data

The results of the NICAM/Joint-Simulator simulation data are shown in Figs. 15 (shallow warm clouds) and 16 (deep convective clouds). Panel (a) in the two figures shows the radiance at 0.659 µm (band 1 of MSI) and brightness temperature at 10.8 µm (band 6 of MSI), respectively. These two panels show the approximate location of target clouds by marking areas with relatively high radiance (yellow-red areas in panel (a) in Fig. 15) and relatively low brightness temperature (blue areas in panel (a) in Fig. 16), respectively.

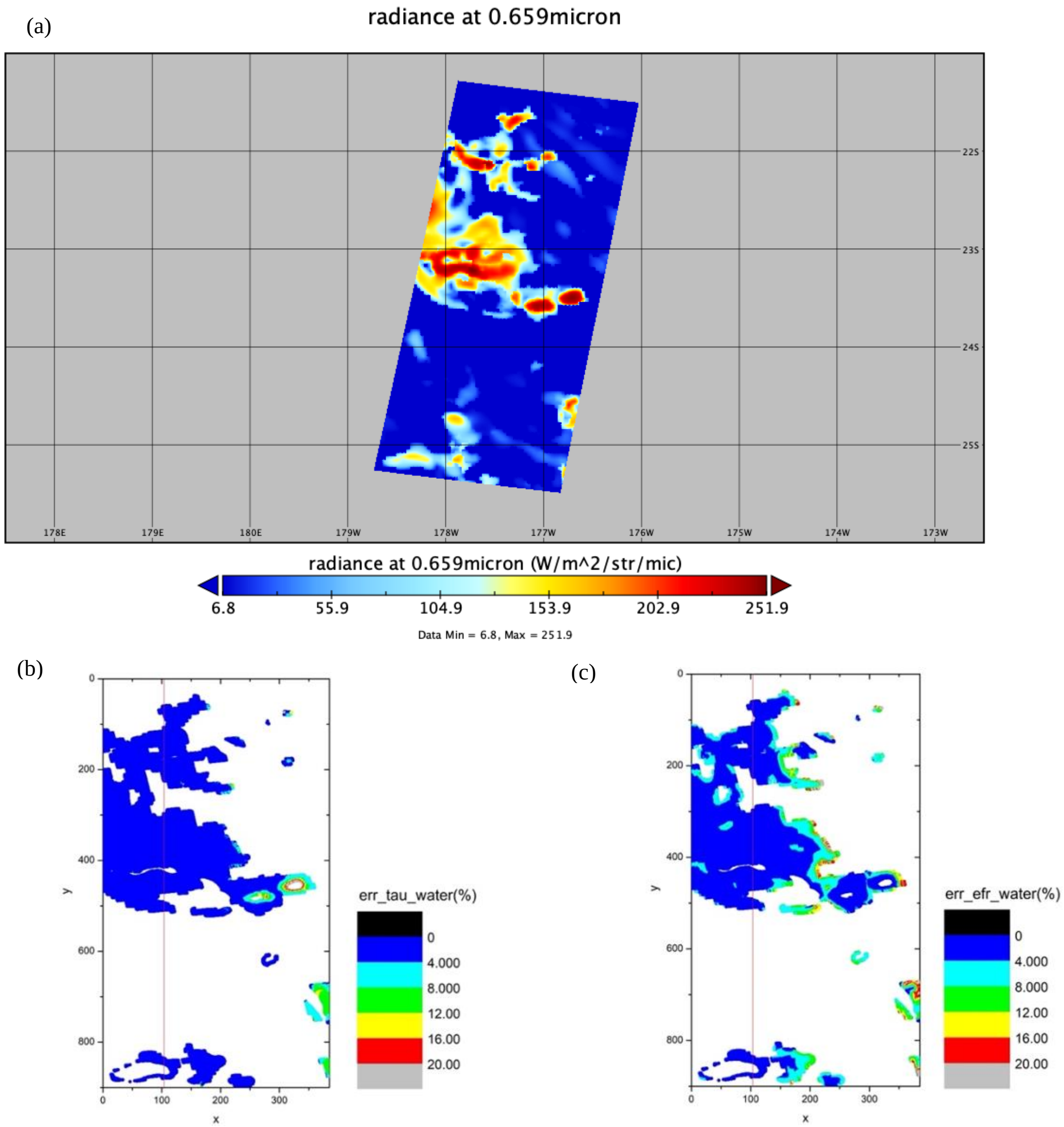

**Figure 15: NICAM/Joint-Simulator data for shallow warm clouds. (a) Radiance at 0.659 µm (band 1 of MSI), (b) error distribution of COT, and (c) error distribution of CDR. X and Y in (b) and (c) show the pixel number in the swath or track direction. The red line in (b) and (c) stands for the location of nadir. (change the "tau" "efr" to COT CDR) (boundary area = convective zone of cloud and clear sky)**

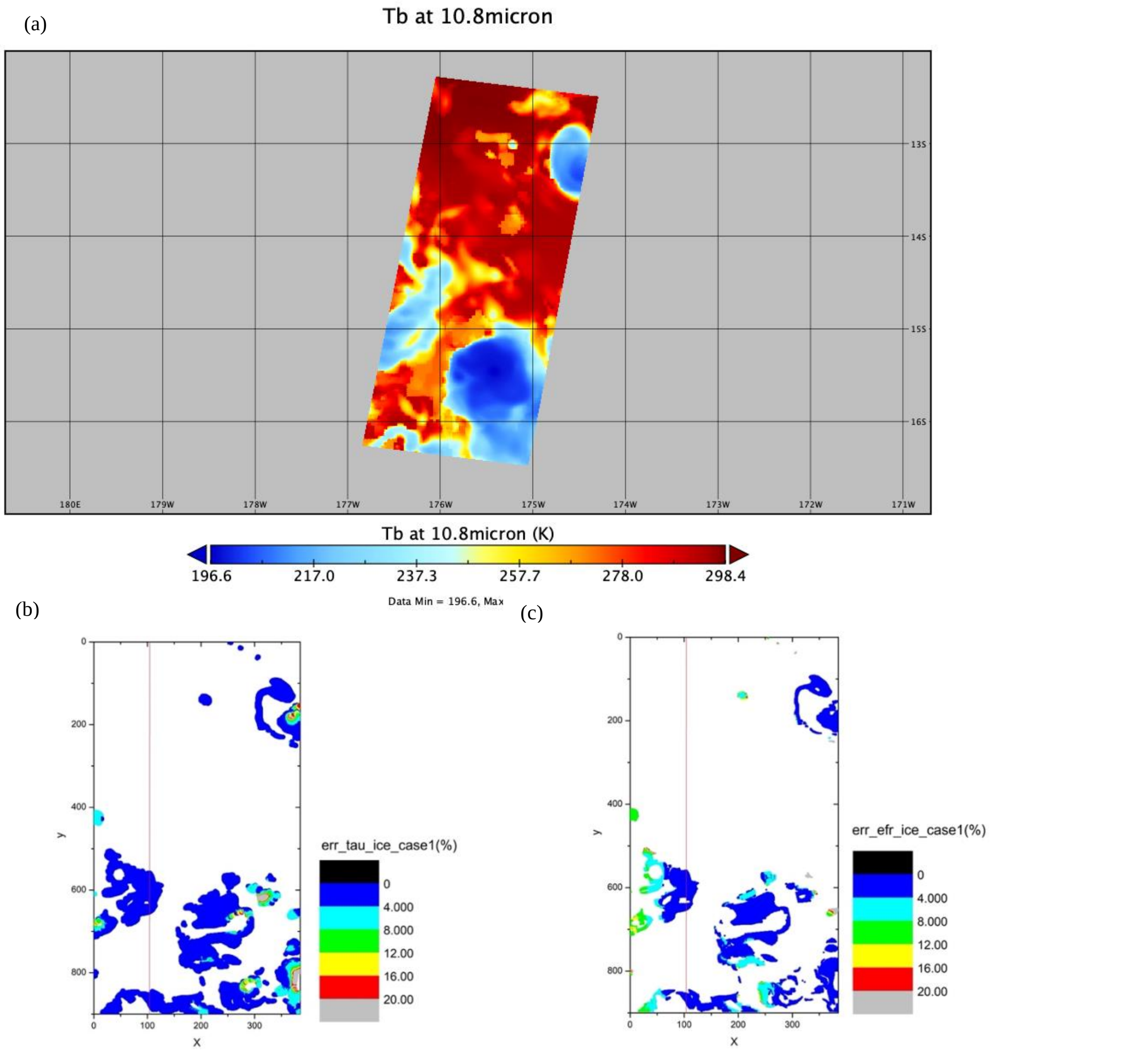

**Figure 16: NICAM/Joint-Simulator data for deep convective clouds. (a) Brightness temperature at 10.8 μm (band 6 of MSI), (b) error distribution of COT, and (c) error distribution of CDR. X and Y in (b) and (c) show the pixel number in the swath or track direction. The red line in (b) and (c) stands for the location of nadir.**

For shallow warm clouds, 71,870 of the 344,064 pixels were defined as water clouds by the MSI cloud profiling algorithm. The average error of COT was 0.89 % and the standard deviation was 1.62 %, whereas the average error of CDR was 3.13 % and the standard deviation was 3.16 %.

For deep convective clouds, 29,501 of the 344,064 pixels were defined as ice cloud by the MSI cloud profiling algorithm. The average error of COT was 1.38 % and the standard deviation was 2.10 %, whereas the average error of CDR was 3.60 % and the standard deviation was 4.17 %.

The spatial distributions of $\Delta\tau$ and $\Delta r_e$ are shown by panels (b) and (c), respectively, in Figs. 15 (shallow warm clouds) and 16 (deep convective clouds). X and Y in panels (b) and (c) indicate the pixel number in the direction of the swath or track. Comparing panel (a) with panels (b) and (c) in each figure showed that the MSI cloud profiling algorithm accurately identified the target clouds for both shallow warm clouds and deep convective clouds, and the shapes of the error distribution areas in panels (b) and (c) and the target cloud area in panel (a) matched well. The value of the response function at the nadir (the $102^{nd}$ pixel) was the same, regardless of the smile effect (Fig. 1). Therefore, our results also showed that $\Delta\tau$ and $\Delta r_e$ were 0 at the $102^{nd}$ pixel, and that the error tended to increase gradually from the nadir toward both sides, which was especially significant for deep convective cloud. Similar to the averaged error, $\Delta r_e$ was larger than $\varDelta\tau$ in the spatial distribution, but in most cases, both $\Delta\tau$ and $\Delta r_e$ were less than 10 % (blue or light blue areas in Fig. 15 and Fig. 16). Although some pixels in panel (c) in Fig. 16 had $\Delta r_e$ larger than 10 %, most of these pixels were the first and last 24 dummy pixels of the swath, meaning that the data from these pixels were unusable for the observations according to EarthCARE/MSI's specification.

Our results from Nakajima-King diagram shown in Fig. 3 to Fig.6 and Table 3 to Table 4, were also matched well with the results from NICAM/Joint-simulator data. As extreme values of smile property on band 1 and band 3 that shown in Table 3 and Table 4, we found that $\varDelta\tau$ on pix_BND1_min and pix_BND1_max was generally larger than it on pix_BND3_min and pix_BND3_max, and when we talked about $\Delta r_e$, the opposite situation was seen. This is basically because band 1 is more sensitive to COT and band 3 is more sensitive to CDR, suggesting that these extreme values of $\varDelta\tau$ (2% ~ 4%) and $\Delta r_e$ (5% ~ 7%) were referenceable during actual observations. Then our results from Synthetic MSI data simulation proved this suggestion well.

Generally, the results of the NICAM/Joint-Simulator data matched those of the CAPCOM radiation transfer simulation well, suggesting that the error in COT and CDR caused by the smile effect was small, and could be regarded as negligible in most cases.

## 4 Conclusions

During the pre-launch phase of EarthCARE, numerous works are settled to reduce errors and ensure accuracy of each observation instrument, in various aspects. As one of them, our work based on both theorical calculation and numerical simulation, providing scientific references for evaluating the influence of smile property as reasonable as possible. Furthermore, since the smile effect could also be seen in other futural optical instruments, our work could also provide some typical examples of how the smile property affects the retrieval of cloud physical quantity, which could be seen as a useful reference for futural cloud observation instruments development.

According to our results for the CAPCOM radiation transfer simulation and observation scene simulation using NICAM/Joint-Simulator data, the Nakajima-King diagrams clearly showed the smile property on four chosen pixel of band 1 and band 3, as

extreme values. Specifically, for typical shallow warm clouds ($\tau$ = 8, $r_e$ = 8 μm), the smile effect on the cloud retrieval was not significant in most cases (up to 6 % error), and for typical deep convective clouds ($\tau$ = 8, $r_e$ = 40 μm), the smile effect on the cloud retrieval was even less significant in most cases (up to 4 % error). Based on the sensitivity of each band to the retrieval of each cloud physical quantity, extreme error of COT and CDR were generally seen in band 1 and band 3, respectively. Furthermore, the Synthetic MSI data provide not only the special distribution of COT / CDR error caused by smile property, but also a significant proof to our results from CAPCOM simulations. Since the result did not exceed our evaluation criteria of 10% in most cases, we suggest that the smile effect does not lead to appreciable errors in cloud retrieval data from EarthCARE/MSI. This study suggests that an onboard correction of the smile properties to the cloud profile algorithm is not necessary for MSI.

However, our simulations in this study are based on observations over oceanic areas, which is much less strongly influenced by surface albedo than land areas. The surface albedo values used in the NICAM/Joint-Simulator data were 0.04–0.05, which did not change substantially throughout the scene. The surface reflectance could be much more complicated in observations over land areas. If the surface reflectance remained constant, it would be sufficient to correct for the albedo radiation. However, because surface reflectance is a function of the observation wavelength, the surface reflectance will affect the cloud retrieval. For the VIS channel (band 1 of MSI), which is close to the red edge of green vegetation, small shifts in the central wavelength can lead to uncertainties due to the rapid change in surface reflectance. Therefore, it requires more work to evaluate the effect of the smile effect during cloud retrievals over land areas, and to determine whether the smile effect is negligible everywhere. Meanwhile, MSI is also used in the works of aerosol retrieval, which can also be affected by the smile effect. Since the smile property on aerosol retrieval is beyond the scope of this study, future works about this evaluation are necessary, too.

Finally, although the special resolution of NICAM (3.5 km) is lower compared with MSI, NICAM has its own advantage of simulating global areas. After focused on two oceanic scenes in this study, we shall set our next task to evaluate the whole orbit, showing the usefulness and potential of the NICAM data for future works.

**Code/Data contributions**

Due to the nature of this research, participants of this study did not agree for their data to be shared publicly, so supporting data is not available.

**Author contributions**

Prof. Nakajima and Prof. Satoh designed the basic frame of simulations, and Dr. Wang carried them out. Prof. Nakajima developed and provided the base model of CAPCOM, which was arranged for this study by Dr. Wang. Dr. Roh provided the synthetic MSI L1 data using NICAM/Joint Simulator. Prof. Suzuki provided the theoretical ground of the analytical estimation of shortwave flux (Session 2.3). Dr. Yoshida provided the wavelength distribution of relative response function on MSI bands

1 to 4 (Fig. 2). Gracious advices and helps from Dr. Kubota during the overall study. Dr. Wang prepared the manuscript with contributions from all co-authors.

**Special thanks**

Special thanks to Dr. Tsuneaki Suzuki (now independent), who did the early part of this study as the predecessor of the corresponding author. Special thanks to ESA for providing the measured value of response functions of EarthCARE/MSI.

**Competing interests**

The authors declare that they have no conflict of interest.

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
