# Peer review of "Evaluation of the Spectral Misalignment (SMILE) on the Earth Clouds, Aerosols and Radiation Explorer (EarthCARE) /Multi-Spectral Imager (MSI) cloud product"

_EGUsphere, 2022_

## Author Response (AR1)

Dear editor and referees:

Thank you for providing these insights. We appreciate the time and effort you and each of the reviewers have dedicated to providing insightful feedback on ways to strengthen our paper, and we will make the corrections according to the comments in the revised manuscript.

The following is a point-by-point response to the specific comments from referee 1:

1. radiation transfer model, or radiative transfer model? Please check which is better. I think the words "radiative transfer model" are better.

RESPONSE: We agreed with your assessment that the words "radiative transfer model" are better. We will change the words in the revised manuscript.

CHANGE: We change the word in L48 of the revised manuscript.

2. L121, the cloud particle size distribution in this paper is

$n(r) = c/r \exp[-(lnr - lnr0)^2 / (2\eth^2)]$

While the Nakajima and Nakajima (1995), the n(r) is:

$n(r) = N / (\sqrt{2pi} \; \eth) \exp[-(lnr - lnr0)^2 / (2\eth^2)]$.

Why your eq. for the first term contains "r", while the Nakajima's is "N"?

Nakajima, T.Y., Nakajima, T., 1995. Wide-Area Determination of Cloud Microphysical Properties from NOAA AVHRR Measurements for FIRE and ASTEX Regions. Journal of the Atmospheric Sciences.

RESPONSE: The "N" in Nakajima and Nakajima (1995) means total particle number, which is an arbitrary constant.

While the equation used in L121 of our manuscript is originally based on Nakajima and King (1990), in which "C" is a constant.

Both equations show the same lognormal distribution function for cloud particle size distribution.

Of course, the particle size distribution is considered in the calculation of the COT in CAPCOM. However, the particle size distribution is just used as a relative value to perceive the frequency dependence of the optical thickness. The COT is not directly calculated from the particle size distribution.

Nakajima, T. and King, M. D.: Determination of the optical thickness and effective particle radius of clouds from reflected solar radiation measurements. Part I: Theory, J. Atmos. Sci., 47, 1878–1893, doi:10.1175/1520- 0469(1990)047<1878:DOTOTA>2.0.CO;2, 1990.

CHANGE: Explanation was added in L128-130 of revised manuscript.

3. L160, the reciprocal of dL/ð  ð  is ð  ð  /dL, not dt/dL. Please check it.

RESPONSE: We have checked the line and confirmed that dt/dL was a mistype. We will fix it in the revised manuscript.

CHANGE: We fix the word in L165 of the revised manuscript.

4. L195, where is the Fsw from? Please give the reference. And what's the S0, n, k mean in Eq. (5)-(11)?

RESPONSE: Eq. (5) is used to represent a theoretical relationship between shortwave radiation (Fsw), solar constant (S0), cloud cover (n), and the change of cloud albedo (Δα).

Since the optical thickness of the gas-only atmosphere is approximately 0.2, the changes in global mean shortwave radiation according to Δα can be expressed as Eq. (5).

Eq. (9) is also a theoretical relationship that can be found in Brenguier et al. 2011, and "k" equals to 3/2.

We will add the explanations about S0, n, k as well as the reference to the revised manuscript.

Brenguier, J. -L., Burnet, F., and Geoffroy, O.: Cloud optical thickness and liquid water path – does the k coefficient vary with droplet concentration? Atmospheric Chemistry and Physics, 11, 9771-9786, doi:10.5194/acp-11-9771-2011, 2011.

CHANGE: Explanations are added in L202-204, L211, and the new reference is added.

5. L209, Eq. (13) presents the relation between Fsw and CDR, if CDR decreased by 10%, Fsw would decrease by about 4.2Wm-2. So, what the relationship between Fsw and COT? I want to know that how COT changes, resulting Fsw changes?

RESPONSE: From Eq. (11) in L204 we can know that when W is a constant ($\Delta W = 0$), then

$$\frac{\Delta \tau}{\tau} = - \frac{\Delta r_e}{r_e},$$

and we can rewrite Eq. (13) as

$$F_{SW} = 343 \times 0.6 \times 0.81 \times 0.51 \times (1 - 0.51) \times \frac{-\Delta r_e}{r_e} = -42 \times \frac{\Delta \tau}{\tau}.$$

So, if COT increased by 10%, then Fsw would decrease by about 4.2 W/m$^2$.

CHANGE: Information was added in L222-227 of revised manuscript.

6. The CAPCOM can used for retrieval of COT, CER, and CTT or CTH. The authors investigated

the smile error on COT and CER, how about the CTH?

RESPONSE: In CAPCOM, CTH is determined by comparing CTT with temperature vertical profile T(z), which is from global objective analysis data. Therefore, the error of CTH is ascribed to the error of CTT, directly.

Since this paper centers on discussing the smile effect on COT and CDR, we did not talk much about CTH or CTT. We believe that the error in CTH (CTT) is expected to be small, at least to have little effect on the shortwave radiation budget. This is because CTT is related to the emissivity determined by the cloud characteristics, and the emissivity does not fluctuate so much, so we believe that the smile effect does not affect the CTT very much.

CHANGE: Information about CTH and CTT is added in L407-412 of the revised manuscript.

The following is our response to the specific comments from referee 2:

Line 100:

"sunny" does not necessarily imply clear-sky. Therefore, I propose to replace the term "sunny" with the term "cloud-free" or "clear-sky".

RESPONSE: We agreed with your assessment. We will replace the term "sunny" with the term "clear-sky" in the revised manuscript.

CHANGE: The term is replaced in L102 of revised manuscript.

Line 137, Table 2:

I assume the "393" in column D refers to the 393 km altitude of the mission orbit? I suggest to add this information in the Table caption or better to add it in Table 1 of the general mission characteristics.

RESPONSE: Yes, the "393" in column D does refer to the 393 km altitude of the mission orbit. We will add this information directly in Table 1.

CHANGE: New line of mission orbit altitude is added in Table 1.

Line 196, 198, equations (5) and (6):

Add an explanation what $S_0$, n and g are

Line 209, equation (13):

The unit for FSW is missing. Please add.

RESPONSE: We will add the explanation and the unit for Fsw in the revised manuscript.

CHANGE: Explanations and the unit for Fsw was added in L202-204, L216, L219 and L225.

Line 336, Figure 15:

There seem to be regions in the error distribution plots (b) and (c) at positions around x=20-100 and

y=850-900 as well as x=350 and y=450 where no error is found but clouds are present according to panel (a). Does this mean that these regions are not shallow warm clouds or does that mean that the error is off the scale? A short explanation would be appreciated.

RESPONSE: The region around x=20-100 and y=850-900 is not defined as shallow warm clouds, while the error is off the scale in the region around x=350 and y=450. We will add a short explanation about these two regions.

CHANGE: Explanation added in L368-370 of revised manuscript.

Line 363-365:

The structure of this sentence is confusing. Please try to reformulate. Also, the statement that delta tau on pix_BND1_min and pix_BND1_max are generally larger than on pix_BND3_min and pix_BND3_max seems contradictory to Tables 3 and 4. From there I read that the error pix_BND1_min > pix_BND3_min but pix_BND1_max < pix_BND3_max and vice versa for re. Please clarify.

RESPONSE: We will reformulate the sentence here according to every case in Table 3 and 4, both for COT error and CDR error.

CHANGE: Reformulated sentences in L381-385 of revised manuscript.

Line 384:

What does "extreme" error mean here? Please clarify or quantify.

RESPONSE: The extreme error can be found from Table 3 and 4 for both COT error and CDR error, we will add an explanation to clarify the exact value of them in the sentence.

CHANGE: explanation added in L402 of revised manuscript.

Line 388-389:

I would suggest to add here that this statement is only true for the water surfaces that were analyzed in this study. As indicated later on, the effect for scenes over land are not quantified yet and therefore the statement that an onboard correction is generally not necessary would probably require an analysis of the land cases too.

Line 390-397:

It is very good that the authors have pointed out that the impact of the smile effect for scenes over land might be much more difficult to quantify and will require more work. It should therefore be made clear in the abstract that the basic conclusion, i.e. that the impact of the smile effect is negligible, is true for water surfaces but needs to be investigated further for land surfaces.

RESPONSE: We will add the explanation in both the conclusion part and the abstract part to state that further works are still needed to analysis the land cases.

CHANGE: Further explanation is added in abstract part.